# WANDERING WITHIN A WORLD:
# ONLINE CONTEXTUALIZED FEW-SHOT LEARNING

**Mengye Ren**[1,3]   **Michael L. Iuzzolino**[2]   **Michael C. Mozer**[2,4]   **Richard S. Zemel**[1,3,5]

[1]University of Toronto   [2]Google Research   [3]Vector Institute
[4]University of Colorado, Boulder   [5]CIFAR
`{mren,zemel}@cs.toronto.edu`
`{michael.iuzzolino,mozer}@colorado.edu`

## ABSTRACT

We aim to bridge the gap between typical human and machine-learning environments by extending the standard framework of few-shot learning to an online, continual setting. In this setting, episodes do not have separate training and testing phases, and instead models are evaluated online while learning novel classes. As in the real world, where the presence of spatiotemporal context helps us retrieve learned skills in the past, our online few-shot learning setting also features an underlying context that changes throughout time. Object classes are correlated within a context and inferring the correct context can lead to better performance. Building upon this setting, we propose a new few-shot learning dataset based on large scale indoor imagery that mimics the visual experience of an agent wandering within a world. Furthermore, we convert popular few-shot learning approaches into online versions and we also propose a new *contextual prototypical memory* model that can make use of spatiotemporal contextual information from the recent past. [1]

## 1   INTRODUCTION

In machine learning, many paradigms exist for training and evaluating models: standard train-then-evaluate, few-shot learning, incremental learning, continual learning, and so forth. None of these paradigms well approximates the naturalistic conditions that humans and artificial agents encounter as they wander within a physical environment. Consider, for example, learning and remembering peoples' names in the course of daily life. We tend to see people in a given environment—work, home, gym, etc. We tend to repeatedly revisit those environments, with different environment base rates, nonuniform environment transition probabilities, and nonuniform base rates of encountering a given person in a given environment. We need to recognize when we do not know a person, and we need to learn to recognize them the next time we encounter them. We are not always provided with a name, but we can learn in a semi-supervised manner. And every training trial is itself an evaluation trial as we repeatedly use existing knowledge and acquire new knowledge. In this article, we propose a novel paradigm, *online contextualized few-shot learning*, that approximates these naturalistic conditions, and we develop deep-learning architectures well suited for this paradigm.

In traditional few-shot learning (FSL) (Lake et al., 2015; Vinyals et al., 2016), training is episodic. Within an isolated episode, a set of new classes is introduced with a limited number of labeled examples per class—the *support* set—followed by evaluation on an unlabeled *query* set. While this setup has inspired the development of a multitude of meta-learning algorithms which can be trained to rapidly learn novel classes with a few labeled examples, the algorithms are focused solely on the few classes introduced in the current episode; the classes learned are not carried over to future episodes. Although incremental learning and continual learning methods (Rebuffi et al., 2017; Hou et al., 2019) address the case where classes are carried over, the episodic construction of these frameworks seems artificial: in our daily lives, we do not learn new objects by grouping them with five other new objects, process them together, and then move on.

---

[1]Our code and dataset are released at: `https://github.com/renmengye/oc-fewshot-public`

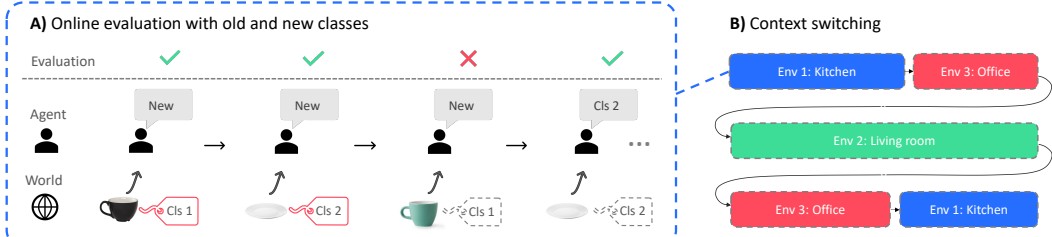

Figure 1: **Online contextualized few-shot learning. A)** Our setup is similar to online learning, where there is no separate testing phase; model training and evaluation happen *at the same time*. The input at each time step is an (image, class-label) pair. The number of classes grows *incrementally* and the agent is expected to answer "new" for items that have not yet been assigned labels. Sequences can be *semi-supervised*; here the label is not revealed for every input item (labeled/unlabeled shown by red solid/grey dotted boxes). The agent is evaluated on the correctness of all answers. The model obtains learning signals only on labeled instances, and is correct if it predicts the label of previously-seen classes, or 'new' for new ones. **B)** The overall sequence switches between different *learning environments*. While the environment ID is *hidden* from the agent, inferring the current environment can help solve the task.

To break the rigid, artificial structure of continual and few-shot learning, we propose a new continual few-shot learning setting where environments are revisited and the total number of novel object classes increases over time. Crucially, model evaluation happens on each trial, very much like the setup in online learning. When encountering a new class, the learning algorithm is expected to indicate that the class is "new," and it is then expected to recognize subsequent instances of the class once a label has been provided.

When learning continually in such a dynamic environment, contextual information can guide learning and remembering. Any structured sequence provides *temporal context*: the instances encountered recently are predictive of instances to be encountered next. In natural environments, *spatial context*— information in the current input weakly correlated with the occurrence of a particular class—can be beneficial for retrieval as well. For example, we tend to see our boss in an office setting, not in a bedroom setting. Human memory retrieval benefits from both spatial and temporal context (Howard, 2017; Kahana, 2012). In our online few-shot learning setting, we provide spatial context in the presentation of each instance and temporal structure to sequences, enabling an agent to learn from both spatial and temporal context. Besides developing and experimenting on a toy benchmark using handwritten characters (Lake et al., 2015), we also propose a new large-scale benchmark for online contextualized few-shot learning derived from indoor panoramic imagery (Chang et al., 2017). In the toy benchmark, temporal context can be defined by the co-occurrence of character classes. In the indoor environment, the context—temporal and spatial—is a natural by-product as the agent wandering in between different rooms.

We propose a model that can exploit contextual information, called *contextual prototypical memory* (*CPM*), which incorporates an RNN to encode contextual information and a separate prototype memory to remember previously learned classes (see Figure 4). This model obtains significant gains on few-shot classification performance compared to models that do not retain a memory of the recent past. We compare to classic few-shot algorithms extended to an online setting, and CPM consistently achieves the best performance.

The main contributions of this paper are as follows. First, we define an *online contextualized few-shot learning (OC-FSL)* setting to mimic naturalistic human learning. Second, we build three datasets: 1) *RoamingOmniglot* is based on handwritten characters from Omniglot (Lake et al., 2015); 2) *RoamingImageNet* is based on images from ImageNet (Russakovsky et al., 2015); and 3) *RoamingRooms* is our new few-shot learning dataset based on indoor imagery (Chang et al., 2017), which resembles the visual experience of a wandering agent. Third, we benchmark classic FSL methods and also explore our CPM model, which combines the strengths of RNNs for modeling temporal context and Prototypical Networks (Snell et al., 2017) for memory consolidation and rapid learning.

## 2 RELATED WORK

In this section, we briefly review paradigms that have been used for few-shot learning (FSL) and continual learning (CL).

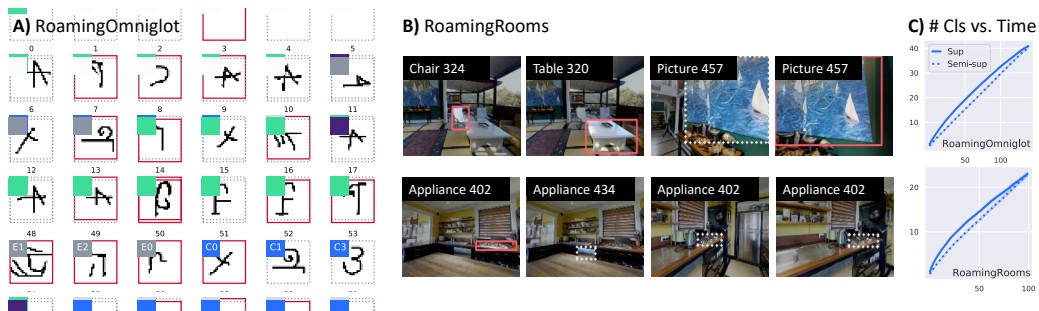

Figure 2: **Sample online contextualized few-shot learning sequences. A)** RoamingOmniglot. Red solid boxes denote labeled examples of Omniglot handwritten characters, and dotted boxes denote unlabeled ones. Environments are shown in colored labels in the top left corner. **B)** Image frame samples of a few-shot learning sequence in our RoamingRooms dataset collected from a random walking agent. The task here is to recognize and classify novel instance IDs in the home environment. Here the groundtruth instance masks/bounding boxes are provided. **C)** The growth of total number of labeled classes in a sequence for RoamingOmniglot (top) and RoamingRooms (bottom).

**Few-shot learning:** FSL (Lake et al., 2015; Li et al., 2007; Koch et al., 2015; Vinyals et al., 2016) considers learning new tasks with few labeled examples. FSL models can be categorized as based on: metric learning (Vinyals et al., 2016; Snell et al., 2017), memory (Santoro et al., 2016), and gradient adaptation (Finn et al., 2017; Li et al., 2017). The model we propose, CPM, lies on the boundary between these approaches, as we use an RNN to model the temporal context but we also use metric-learning mechanisms and objectives to train.

Several previous efforts have aimed to extend few-shot learning to incorporate more natural constraints. One such example is semi-supervised FSL (Ren et al., 2018), where models also learn from a pool of unlabeled examples. While traditional FSL only tests the learner on novel classes, *incremental FSL* (Gidaris & Komodakis, 2018; Ren et al., 2019) tests on novel classes together with a set of base classes. These approaches, however, have not explored how to iteratively add new classes.

In concurrent work, Antoniou et al. (2020) extend FSL to a continual setting based on image sequences, each of which is divided into stages with a fixed number of examples per class followed by a query set. It focuses on more flexible and faster adaptation since the models are evaluated online, and the context is a soft constraint instead of a hard separation of tasks. Moreover, new classes need to be identified as part of the sequence, crucial to any learner's incremental acquisition of knowledge.

**Continual learning:** Continual (or lifelong) learning is a parallel line of research that aims to handle a sequence of dynamic tasks (Kirkpatrick et al., 2017; Li & Hoiem, 2018; Lopez-Paz & Ranzato, 2017; Yoon et al., 2018). A key challenge here is catastrophic forgetting (McCloskey & Cohen, 1989), where the model "forgets" a task that has been learned in the past. Incremental learning (Rebuffi et al., 2017; Castro et al., 2018; Wu et al., 2019; Hou et al., 2019; He et al., 2020) is a form of continual learning, where each task is an incremental batch of several new classes. This assumption that novel classes always come in batches seems unnatural.

Traditionally, continual learning is studied with tasks such as permuted MNIST (Lecun et al., 1998) or split-CIFAR (Krizhevsky, 2009). Recent datasets aim to consider more realistic continual learning, such as CORe50 (Lomonaco & Maltoni, 2017) and OpenLORIS (She et al., 2019). We summarize core features of these continual learning datasets in Appendix A. First, both CORe50 and OpenLORIS have relatively few object classes, which makes meta-learning approaches inapplicable; second, both contain images of small objects with minimal occlusion and viewpoint changes; and third, OpenLORIS does not have the desired incremental class learning.

In concurrent work, Caccia et al. (2020) proposes a setup to unify continual learning and meta-learning with a similar online evaluation procedure. However, there are several notable differences. First, their models focus on a general loss function without a specific design for predicting new classes; they predict new tasks by examining if the loss of exceeds some threshold. Second, the sequences of inputs are fully supervised. Lastly, their benchmarks are based on synthetic task sequences such as Omniglot or tiered-ImageNet, which are less naturalistic than our RoamingRooms dataset.

**Online meta-learning:** Some existing work builds on early approaches (Thrun, 1998; Schmidhuber, 1987) that tackle continual learning from a meta-learning perspective. Finn et al. (2019) proposes storing all task data in a data buffer; by contrast, Javed & White (2019) proposes to instead learn a good representation that supports such online updates. In Jerfel et al. (2019), a hierarchical Bayesian mixture model is used to address the dynamic nature of continual learning.

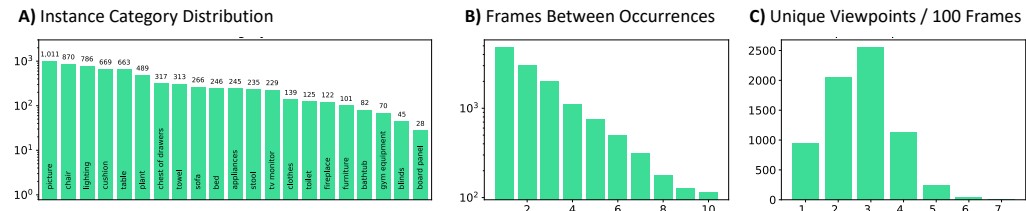

Figure 3: **Statistics for our RoamingRooms dataset.** Plots show a natural long tail distribution of instances grouped into categories. An average sequence has 3 different view points. Sequences are highly correlated in time but revisits are not uncommon.

**Connections to the human brain:** Our CPM model consists of multiple memory systems, consistent with claims of cognitive neuroscientists of multiple memory systems in the brain. The complementary learning systems (CLS) theory (McClelland et al., 1995) suggests that the hippocampus stores the recent experience and is likely where few-shot learning takes place. However, our model is more closely related to contextual binding theory (Yonelinas et al., 2019), which suggests that long-term encoding of information depends on binding spatiotemporal context, and without this context as a cue, forgetting occurs. Our proposed CPM contains parallels to human brain memory components (Cohen & Squire, 1980). Long term statistical learning is captured in a CNN that produces a deep embedding. An RNN holds a type of working memory that can retain novel objects and spatiotemporal contexts. Lastly, the prototype memory represents the semantic memory, which consolidates multiple events into a single knowledge vector (Duff et al., 2020). Other deep learning researchers have proposed multiple memory systems for continual learning. In Parisi et al. (2018), the learning algorithm is heuristic and representations come from pretrained networks. In Kemker & Kanan (2018), a prototype memory is used for recalling recent examples and rehearsal from a generative model allows this knowledge to be integrated and distilled into a long-term memory.

## 3 ONLINE CONTEXTUALIZED FEW-SHOT LEARNING

In this section, we introduce our new online contextualized few-shot learning (OC-FSL) setup in the form of a sequential decision problem, and then introduce our new benchmark datasets.

**Continual few-shot classification as a sequential decision problem:** In traditional few-shot learning, an episode is constructed by a support set $S$ and a query set $Q$. A few-shot learner $f$ is expected to predict the class of each example in the query set $\mathbf{x}^Q$ based on the support set information: $\hat{y}^Q = f(\mathbf{x}^Q; (\mathbf{x}_1^S, y_1^S), \dots, (\mathbf{x}_N^S, y_N^S))$, where $\mathbf{x} \in \mathbb{R}^D$, and $y \in [1 \dots K]$. This setup is not a natural fit for continual learning, since it is unclear when to insert a query set into the sequence.

Inspired by the online learning literature, we can convert continual few-shot learning into a sequential decision problem, where every input example is also part of the evaluation: $\hat{y}_t = f(\mathbf{x}_t; (\mathbf{x}_1, \tilde{y}_1), \dots, (\mathbf{x}_{t-1}, \tilde{y}_{t-1}))$, for $t = 1 \dots T$, where $\tilde{y}$ here further allows that the sequence of inputs to be semi-supervised: $\tilde{y}$ equals $y_t$ if labeled, or otherwise $-1$. The setup in Santoro et al. (2016) and Kaiser et al. (2017) is similar; they train RNNs using such a temporal sequence as input. However, their evaluation relies on a "query set" at the end. We instead evaluate online while learning.

Figure 1-A illustrates these features, using an example of an input sequence where an agent is learning about new objects in a kitchen. The model is rewarded when it correctly predicts a known class or when it indicates that the item has yet to be assigned a label.

**Contextualized environments:** Typical continual learning consists of a sequence of tasks, and models are trained sequentially for each task. This feature is also preserved in many incremental learning settings (Rebuffi et al., 2017). For instance, the split-CIFAR task divides CIFAR-100 into 10 learning environments, each with 10 classes, presented sequentially. In our formulation, the sequence returns to earlier environments (see Figure 1-B), which enables assessment of long-term durability of knowledge. Although the ground-truth environment identity is not provided, we structure the task such that the environment itself provides contextual cues which can constrain the correct class label. *Spatial* cues in the input help distinguish one environment from another. *Temporal* cues are implicit because the sequence tends to switch environments infrequently, allowing recent inputs to be beneficial in guiding the interpretation of the current input.

**RoamingOmniglot:** The Omniglot dataset (Lake et al., 2015) contains 1623 handwritten characters from 50 different alphabets. We split the alphabets into 31 for training, 5 for validation, and 13 for testing. We augment classes by 90 degree rotations to create 6492 classes in total. Each contextualized few-shot learning image sequence contains 150 images, drawn from a random sample

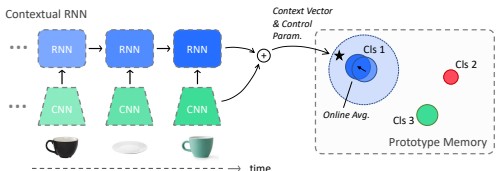

Figure 4: **Contextual prototypical memory.** Temporal contextual features are extracted from an RNN. The prototype memory stores one vector per class and does online averaging. Examples falling outside the radii of all prototypes are classified as "new."

of 5-10 alphabets, for a total of 50 classes per sequence. These classes are randomly assigned to 5 different environments; within an environment, the characters are distributed according to a Chinese restaurant process (Aldous, 1985) to mimic the imbalanced long-tail distribution of naturally occurring objects. We switch between environments using a Markov switching process; i.e., at each step there is a constant probability of switching to another environment. An example sequence is shown in Figure 2-A.

**RoamingRooms:** As none of the current few-shot learning datasets provides the natural online learning experience that we would like to study, we created our own dataset using simulated indoor environments. We formulate this as a few-shot instance learning problem, which could be a use case for a home robot: it needs to quickly recognize and differentiate novel object instances, and large viewpoint variations can make this task challenging (see examples in Figure 2-B). There are over 7,000 unique instance classes in the dataset, making it suitable to meta-learning approaches.

Our dataset is derived from the Matterport3D dataset (Chang et al., 2017) with 90 indoor worlds captured using panoramic depth cameras. We split these into 60 worlds for training, 10 for validation and 20 for testing. MatterSim (Anderson et al., 2018) is used to load the simulated world and collect camera images and HabitatSim (Savva et al., 2019) is used to project instance segmentation labels onto 2D image space. We created a random walking agent to collect the virtual visual experience. For each viewpoint in the random walk, we randomly sample one object from the image sensor and highlight it with the available instance segmentation, forming an input *frame*. Each viewpoint provides environmental context—the other objects present in the room with the highlighted object.

Figure 3-A shows the object instance distribution. We see strong temporal correlation, as 30% of the time the same instance appears in the next frame (Figure 3-B), but there is also a significant proportion of revisits. On average, there are three different viewpoints per 100-image sequence (Figure 3-C). Details and other statistics of our proposed datasets are included in the Appendix.

**RoamingImageNet:** To make the classification problem more challenging, we also report results on RoamingImageNet, which uses the same online sequence sampler as RoamingOmniglot but applied on the *tiered*-ImageNet dataset (Ren et al., 2018), a subset of the original ImageNet dataset (Russakovsky et al., 2015). It contains 608 classes with a total of 779K images of size 84×84. We use the same split of classes as Ren et al. (2018). The high-level categories are used for sampling different "environments" just like the notion of alphabets in RoamingOmniglot.

## 4 CONTEXTUAL PROTOTYPICAL MEMORY NETWORKS

In the online contextualized few-shot learning setup, the few-shot learner can potentially improve by modeling the temporal context. Metric learning approaches (Snell et al., 2017) typically ignore temporal contextual relations and directly compare the similarity between training and test samples. Gradient-based approaches (Javed & White, 2019), on the other hand, have the ability to adapt to new contexts, but they do not naturally handle new and unlabeled examples. Inspired by the contextual binding theory of human memory (Yonelinas et al., 2019), we propose a simple yet effective approach that uses an RNN to transmit spatiotemporal context and control signals to a prototype memory (Figure 4).

**Prototype memory:** We start describing our model with the prototype memory, which is an online version of the Prototypical Network (or *ProtoNet*) (Snell et al., 2017). ProtoNet can be viewed as a knowledge base memory, where each object class $k$ is represented by a prototype vector $\mathbf{p}[k]$, computed as the mean vector of all the support instances of the class in a sequence. It can also be applied to our task of online few-shot learning naturally, with some modifications. Suppose that at time-step $t$ we have already stored a few classes in the memory, each represented by their current prototype $\mathbf{p}_t[k]$, and we would like to query the memory using the input feature $\mathbf{h}_t$. We model our prototype memory as $\hat{y}_{t,k} = \mathrm{softmax}(-d_{\mathbf{m}_t}(\mathbf{h}_t, \mathbf{p}_t[k]))$, (e.g. squared Euclidean distance or cosine dissimilarity) parameterized by a vector $\mathbf{m}_t$ that scales each dimension with a Hadamard product. To predict whether an example is of a new class, we can use a separate *novelty* output $\hat{u}_t^r$

Table 1: **RoamingOmniglot OC-FSL results.** Max 5 env, 150 images, 50 cls, with $8\times8$ occlusion.

| Method | Supervised | | | Semi-supervised | | |
|---|---|---|---|---|---|---|
| | AP | 1-shot Acc. | 3-shot Acc. | AP | 1-shot Acc. | 3-shot Acc. |
| LSTM | 64.34 | $61.00 \pm 0.22$ | $81.85 \pm 0.21$ | 54.34 | $68.30 \pm 0.20$ | $76.38 \pm 0.49$ |
| DNC | 81.30 | $78.87 \pm 0.19$ | $91.01 \pm 0.15$ | 81.37 | $88.56 \pm 0.12$ | $93.81 \pm 0.26$ |
| OML-U | 77.38 | $70.98 \pm 0.21$ | $89.13 \pm 0.16$ | 66.70 | $74.65 \pm 0.19$ | $90.81 \pm 0.34$ |
| OML-U++ | 86.85 | $88.43 \pm 0.14$ | $92.01 \pm 0.14$ | 81.39 | $71.64 \pm 0.19$ | $93.72 \pm 0.27$ |
| O-MN | 88.69 | $84.82 \pm 0.15$ | $95.55 \pm 0.11$ | 84.39 | $88.77 \pm 0.13$ | $97.28 \pm 0.17$ |
| O-IMP | 90.15 | $85.74 \pm 0.15$ | $96.66 \pm 0.09$ | 81.62 | $88.68 \pm 0.13$ | $97.09 \pm 0.19$ |
| O-PN | 90.49 | $85.68 \pm 0.15$ | $96.95 \pm 0.09$ | 84.61 | $88.71 \pm 0.13$ | $97.61 \pm 0.17$ |
| CPM (Ours) | **94.17** | $\mathbf{91.99} \pm 0.11$ | $\mathbf{97.74} \pm 0.08$ | **90.42** | $\mathbf{93.18} \pm 0.16$ | $\mathbf{97.89} \pm 0.15$ |

Table 2: **RoamingRooms OC-FSL results.** Max 100 images and 40 classes.

| Method | Supervised | | | Semi-supervised | | |
|---|---|---|---|---|---|---|
| | AP | 1-shot Acc. | 3-shot Acc. | AP | 1-shot Acc. | 3-shot Acc. |
| LSTM | 45.67 | $59.90 \pm 0.40$ | $61.85 \pm 0.45$ | 33.32 | $52.71 \pm 0.38$ | $55.83 \pm 0.76$ |
| DNC | 80.86 | $82.15 \pm 0.32$ | $87.30 \pm 0.30$ | 73.49 | $80.27 \pm 0.33$ | $87.87 \pm 0.49$ |
| OML-U | 76.27 | $73.91 \pm 0.37$ | $83.99 \pm 0.33$ | 63.40 | $70.67 \pm 0.38$ | $85.25 \pm 0.56$ |
| OML-U++ | 88.03 | $88.32 \pm 0.27$ | $89.61 \pm 0.29$ | 81.90 | $84.79 \pm 0.31$ | $89.80 \pm 0.47$ |
| O-MN | 85.91 | $82.82 \pm 0.32$ | $89.99 \pm 0.26$ | 78.99 | $80.08 \pm 0.34$ | $\mathbf{92.43} \pm 0.41$ |
| O-IMP | 87.33 | $85.28 \pm 0.31$ | $90.83 \pm 0.25$ | 75.36 | $84.57 \pm 0.31$ | $91.17 \pm 0.43$ |
| O-PN | 86.01 | $84.89 \pm 0.31$ | $89.58 \pm 0.28$ | 76.36 | $80.67 \pm 0.34$ | $88.83 \pm 0.49$ |
| CPM (Ours) | **89.14** | $\mathbf{88.39} \pm 0.27$ | $\mathbf{91.31} \pm 0.26$ | **84.12** | $\mathbf{86.17} \pm 0.30$ | $91.16 \pm 0.44$ |

with sigmoid activation, similar to the approach introduced in Ren et al. (2018), where $\beta_t^r$ and $\gamma_t^r$ are yet-to-be-specified thresholding hyperparameters (the superscript $r$ stands for read):

$$\hat{u}_t^r = \sigma((\min_k d_{\mathbf{m}_t}(\mathbf{h}_t, \mathbf{p}_t[k]) - \beta_t^r)/\gamma_t^r). \tag{1}$$

**Memory consolidation with online prototype averaging:** Traditionally, ProtoNet uses the average representation of a class across all support examples. Here, we must be able to adapt the prototype memory incrementally at each step. Fortunately, we can recover the computation of a ProtoNet by performing a simple online averaging:

$$c_t = c_{t-1} + 1; \quad A(\mathbf{h}_t; \mathbf{p}_{t-1}, c_t) = \frac{1}{c_t}\left(\mathbf{p}_{t-1}c_{t-1} + \mathbf{h}_t\right), \tag{2}$$

where $\mathbf{h}$ is the input feature, and $\mathbf{p}$ is the prototype, and $c$ is a scalar indicating the number of examples that have been added to this prototype up to time $t$. The online averaging function $A$ can also be made more flexible to allow more plasiticity, modeled by a *gated averaging unit* (GAU):

$$A_{\text{GAU}}(\mathbf{h}_t; \mathbf{p}_{t-1}) = (1 - f_t) \cdot \mathbf{p}_{t-1} + f_t \cdot \mathbf{h}_t, \quad \text{where} \quad f_t = \sigma(W_f[\mathbf{h}_t, \mathbf{p}_{t-1}] + b_f) \in \mathbb{R}. \tag{3}$$

When the current example is unlabeled, $\tilde{y}_t$ is encoded as $-1$, and the model's own prediction $\hat{y}_t$ will determine which prototype to update; in this case, the model must also determine a strength of belief, $\hat{u}_t^w$, that the current unlabeled example should be treated as a new class. Given $\hat{u}_t^w$ and $\hat{y}_t$, the model can then update a prototype:

$$\hat{u}_t^w = \sigma((\min_k d_{\mathbf{m}_t}(\mathbf{h}_t, \mathbf{p}_t[k]) - \beta_t^w)/\gamma_t^w), \tag{4}$$

$$\Delta[k]_t = \underbrace{\mathbb{1}[\tilde{y}_t = k]}_{\text{Supervised}} + \underbrace{\hat{y}_{t,k}(1 - \hat{u}_t^w)\mathbb{1}[\tilde{y}_t = -1]}_{\text{Unsupervised}}, \tag{5}$$

$$c[k]_t = c[k]_{t-1} + \Delta[k]_t, \tag{6}$$

$$\mathbf{p}[k]_t = A(\mathbf{h}_t\Delta[k]_t; \mathbf{p}[k]_{t-1}, c[k]_t), \quad \text{or} \quad \mathbf{p}[k]_t = A_{\text{GAU}}(\mathbf{h}_t\Delta[k]_t; \mathbf{p}[k]_{t-1}). \tag{7}$$

As-yet-unspecified hyperparameters $\beta_t^w$ and $\gamma_t^w$ are required (the superscript $w$ is for write). These parameters for the online-updating novelty output $\hat{u}_t^w$ are distinct from $\beta_t^r$ and $\gamma_t^r$ in Equation 1. The intuition is that for "self-teaching" to work, the model potentially needs to be more conservative in creating new classes (avoiding corruption of prototypes) than in predicting an input as being a new class.

**Contextual RNN:** Instead of directly using the features from the CNN $\mathbf{h}_t^{\text{CNN}}$ as input features to the prototype memory, we would also like to use contextual information from the recent past. Above we introduced threshold hyperparameters $\beta_t^r, \gamma_t^r, \beta_t^w, \gamma_t^w$ as well as the metric parameter $\mathbf{M}_t$. We let the contextual RNN output these additional control parameters, so that the unknown thresholds and metric function can adapt based on the information in the context. The RNN produces the context vector $\mathbf{h}_t^{\text{RNN}}$ and other control parameters conditioned on $\mathbf{h}_t^{\text{CNN}}$:

$$\left[\mathbf{z}_t, \mathbf{h}_t^{\text{RNN}}, \mathbf{m}_t, \beta_t^r, \gamma_t^r, \beta_t^w, \gamma_t^w\right] = \text{RNN}(\mathbf{h}_t^{\text{CNN}}; \mathbf{z}_{t-1}), \tag{8}$$

Table 3: **RoamingImageNet OC-FSL results.** Max 150 images and 50 classes. * denotes CNN pretrained using regular classification.

| Method | Supervised | | | Semi-supervised | | |
|--------|------|------------|------------|------|------------|------------|
| | AP | 1-shot Acc. | 3-shot Acc. | AP | 1-shot Acc. | 3-shot Acc. |
| LSTM* | 22.54 | $28.14 \pm 0.20$ | $52.07 \pm 0.27$ | 13.50 | $30.02 \pm 0.20$ | $46.95 \pm 0.56$ |
| DNC* | 26.80 | $33.45 \pm 0.19$ | $55.78 \pm 0.27$ | 16.50 | $39.53 \pm 0.19$ | $54.10 \pm 0.54$ |
| OML-U | 21.89 | $15.06 \pm 0.14$ | $52.52 \pm 0.27$ | 10.16 | $22.74 \pm 0.17$ | $55.81 \pm 0.55$ |
| O-MN | 13.05 | $20.61 \pm 0.15$ | $38.73 \pm 0.24$ | 9.32 | $25.96 \pm 0.16$ | $55.32 \pm 0.51$ |
| O-IMP | 14.25 | $22.92 \pm 0.16$ | $41.01 \pm 0.25$ | 4.55 | $20.70 \pm 0.15$ | $51.23 \pm 0.53$ |
| O-PN* | 23.10 | $32.82 \pm 0.19$ | $49.98 \pm 0.25$ | 15.76 | $36.69 \pm 0.18$ | $55.47 \pm 0.53$ |
| CPM (Ours) | **34.43** | **$40.40 \pm 0.21$** | **$60.29 \pm 0.26$** | **24.75** | **$44.58 \pm 0.21$** | **$58.72 \pm 0.53$** |

where $\mathbf{z}_t$ is the recurrent state of the RNN, and $\mathbf{m}_t$ is the scaling factor in the dissimilarity score. The context, $\mathbf{h}_t^{\text{RNN}}$, serves as an additive bias on the state vector used for FSL: $\mathbf{h}_t = \mathbf{h}_t^{\text{CNN}} + \mathbf{h}_t^{\text{RNN}}$. This addition operation in the feature space can help contextualize prototypes based on temporal proximity, and is also similar to how the human brain leverages spatiotemporal context for memory storage (Yonelinas et al., 2019).

**Loss function:** The loss function is computed after an entire sequence ends and all network parameters are learned end-to-end. The loss is composed of two parts. The first is binary cross-entropy (BCE), for telling whether each example has been assigned a label or not, i.e., prediction of new classes, and $u_t$ is the ground-truth binary label. Second we use a multi-class cross-entropy for classifying among the known ones. We can write down the overall loss function as follows:

$$\mathcal{L} = \frac{1}{T} \sum_{t=1}^{T} \lambda \underbrace{\left[ -u_t \log(\hat{u}_t^r) - (1 - u_t) \log(1 - \hat{u}_t^r) \right]}_{\text{Binary cross entropy on old vs. new}} + \sum_{k=1}^{K} \underbrace{-\mathbb{1}[y_t = k](1 - u_t) \log(\hat{y}_{t,k})}_{\text{Cross entropy on old classes}}. \quad (9)$$

**Training and evaluation:** During training, we sample learning sequences and for each sequence, we perform one iterative update to minimize the loss function (Eq. 9). At the beginning of each sequence, the memory is reset. During training, the model learns from a set of training classes. During test time, the model recognizes new classes that have never been seen during training.

## 5 EXPERIMENTS

In this section, we show experimental results for our online contextualized few-shot learning paradigm, using RoamingOmniglot and RoamingRooms (see Sec. 3) to evaluate our model, CPM, and other state-of-the-art methods. For Omniglot, we apply an $8 \times 8$ CutOut (Devries & Taylor, 2017) to each image to make the task more challenging.

**Implementation details:** For RoamingOmniglot, we use the common 4-layer CNN for few-shot learning with 64 channels in each layer. For RoamingImageNet, we also use ResNet-12 with input resolution $84 \times 84$ (Oreshkin et al., 2018). For the RoamingRooms, we resize the input to $120 \times 160$ and use ResNet-12. To represent the feature of the input image with an attention mask, we concatenate the global average pooled feature with the attention ROI feature, resulting in a 512d feature vector. For the contextual RNN, in both experiments we used an LSTM (Hochreiter & Schmidhuber, 1997) with a 256d hidden state. The best CPM model is equipped using GAU and cosine similarity for querying prototypes. Logits based on cosine similarity are multiplied with a learned scalar initialized at 10.0 (Oreshkin et al., 2018). We include additional training details in Appendix B.

**Evaluation metrics:** In order to compute a single number that characterizes the learning ability over sequences, we propose to use *average precision* (AP) to evaluate both with respect to old versus new and the specific class predictions. Concretely, all predictions are sorted by their old vs. new scores, and we compute AP using the area under the precision-recall curve. A true positive is defined as the correct prediction of a multi-class classification among known classes. We also compute the "$N$-shot" accuracy; i.e., the average accuracy after seeing the label $N$ times in the sequence. Note that these accuracy scores only reflect the performance on *known* class predictions. All numbers are reported with an average over 2,000 sequences and for $N$-shot accuracy standard error is also included. Further explanation of these metrics is in Appendix A.3.

**Comparisons:** To evaluate the merits of our proposed model, we implement classic few-shot learning and online meta-learning methods. More implementation and training details of these baseline methods can be found in Appendix B.

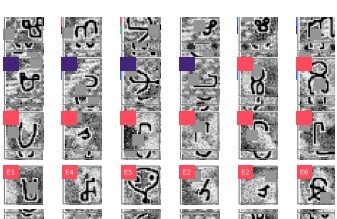

Figure 5: **Few-shot classification accuracy over time. Left:** RoamingOmniglot. **Right:** RoamingRooms. **Top:** Supervised. **Bottom:** Semi-supervised. An offline logistic regression (Offline LR) baseline is also included, using pretrained ProtoNet features. It is trained on all labeled examples except for the one at the current time step.

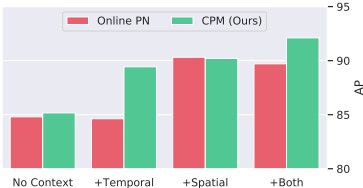

Figure 6: **Effect of spatiotemporal context.** Spatiotemporal context are added separately and together in RoamingOmniglot, by introducing texture background and temporal correlation. **Left:** Stimuli used for spatial cue of the background environment. **Right:** Our CPM model benefits from the presence of a temporal context ("+Temporal" and "+Both")

- **OML** (Javed & White, 2019): This is an online version of MAML (Finn et al., 2017). It performs one gradient descent step for each labeled input image, and slow weights are learned via backpropagation through time. On top of OML, we added an unknown predictor $\hat{u}_t = 1 - \max_k \hat{y}_{t,k}$ [2] (**OML-U**). We also found that using cosine classifier without the last layer ReLU is usually better than using the original dot-product classifier, and this improvement is denoted as **OML-U++**.

- **LSTM** (Hochreiter & Schmidhuber, 1997) & **DNC** (Graves et al., 2016): We include RNN methods for comparison as well. Differentiable neural computer (DNC) is an improved version of memory augmented neural network (MANN) (Santoro et al., 2016).

- **Online MatchingNet (O-MN)** (Vinyals et al., 2016), **IMP (O-IMP)** (Allen et al., 2019) & **ProtoNet (O-PN)** (Snell et al., 2017): We used the same negative Euclidean distance as the similarity function for these three metric learning based approaches. In particular, MatchingNet stores all examples and performs nearest neighbor matching, which can be memory inefficient. Note that Online ProtoNet is a variant of our method without the contextual RNN.

**Main results:** Our main results are shown in Table 1, 2 and 3, including both supervised and semi-supervised settings. Our approach achieves the best performance on AP consistently across all settings. Online ProtoNet is a direct comparison without our contextual RNN and it is clear that CPM is significantly better. Our method is slightly worse than Online MatchingNet in terms of 3-shot accuracy on the RoamingRooms semisupervised benchmark. This can be explained by the fact that MatchingNet stores all past seen examples, whereas CPM only stores one prototype per class. Per timestep accuracy is plotted in Figure 5, and the decaying accuracy is due to the increasing number of classes over time. In RoamingOmniglot, CPM is able to closely match or even sometimes surpass the offline classifier, which re-trains at each step and uses all images in a sequence except the current one. This is reasonable as our model is able to leverage information from the current context.

**Effect of spatiotemporal context:** To answer the question whether the gain in performance is due to spatiotemporal reasoning, we conduct the following experiment comparing CPM with online ProtoNet. We allow the CNN to have the ability to recognize the context in RoamingOmniglot by adding a texture background image using the Kylberg texture dataset (Kylberg, 2011) (see Figure 6 left). As a control, we can also destroy the temporal context by shuffling all the images in a sequence. We train four different models on dataset controls with or without the presence of spatial or temporal context, and results are shown in Figure 6. First, both online ProtoNet and CPM benefit from the inclusion of a spatial context. This is understandable as the CNN has the ability to learn spatial cues, which re-confirms our main hypothesis that successful inference of the current context is beneficial to novel object recognition. Second, only our CPM model benefits from the presence of temporal context, and it receives distinct gains from spatial and temporal contexts.

---

[2]We tried a few other ways and this is found to be the best.

Table 4: **Effect of forgetting over a time interval on RoamingOmniglot.** Average accuracy vs. the number of time steps since the model has last seen the label of a particular class.

| Interval | Supervised | | | | | | Semi-Supervised | | | | | |
|---|---|---|---|---|---|---|---|---|---|---|---|---|
| | 1 - 2 | 3 - 5 | 6 - 10 | 11 - 20 | 21 - 50 | 51 - 100 | 1 - 2 | 3 - 5 | 6 - 10 | 11 - 20 | 21 - 50 | 51 - 100 |
| OPN 1-Shot | 88.8 | 86.9 | 85.2 | 84.7 | 83.6 | 81.1 | 90.1 | 88.9 | 88.4 | 87.6 | 87.3 | 85.1 |
| CPM 1-Shot | **96.1** | **94.0** | **93.0** | **91.6** | **88.2** | **84.6** | **95.9** | **93.8** | **92.8** | **91.8** | **89.4** | **85.7** |
| OPN 3-Shot | 97.2 | 97.1 | 96.6 | 96.7 | **96.5** | 95.3 | 97.8 | 97.3 | 97.1 | **97.8** | **97.7** | **96.8** |
| CPM 3-Shot | **98.5** | **98.2** | **97.5** | **97.2** | 95.4 | **95.5** | **98.7** | **97.5** | **97.5** | 96.5 | 96.3 | 92.9 |

Table 5: Ablation of CPM architectural components on RoamingOmniglot

| Method | $h^{\text{RNN}}$ | $\beta_t^*, \gamma_t^*$ | Metric $m_t$ | GAU | Val AP |
|---|---|---|---|---|---|
| O-PN | | | | | 91.22 |
| No $h^{\text{RNN}}$ | | ✓ | ✓ | | 92.52 |
| $h^{\text{RNN}}$ only | ✓ | | | | 93.48 |
| No metric $m_t$ | ✓ | ✓ | | | 93.61 |
| No $\beta_t^*, \gamma_t^*$ | ✓ | | ✓ | | 93.98 |
| $h_t = h^{\text{RNN}}$ | ✓ | ✓ | ✓ | | 93.70 |
| CPM Avg. Euc | ✓ | ✓ | ✓ | | 94.08 |
| CPM Avg. Cos | ✓ | ✓ | ✓ | | 94.57 |
| CPM GAU Euc | ✓ | ✓ | ✓ | ✓ | 94.11 |
| CPM GAU Cos | ✓ | ✓ | ✓ | ✓ | **94.65** |

Table 6: Ablation of semi-supervised learning components on RoamingOmniglot

| Method | RNN | Prototype | $\beta_t^w, \gamma_t^w$ | GAU | Val AP |
|---|---|---|---|---|---|
| O-PN | | | | | 90.83 |
| O-PN | | ✓ | | | 89.10 |
| O-PN | | ✓ | ✓ | | 91.22 |
| CPM | | | | | 92.57 |
| CPM | ✓ | | | | 93.16 |
| CPM | ✓ | ✓ | | | 93.20 |
| CPM | ✓ | ✓ | ✓ | | 94.08 |
| CPM | ✓ | ✓ | ✓ | ✓ | **94.65** |

**Effect of forgetting:** As the number of learned classes increases, we expect the average accuracy to drop. To further investigate this forgetting effect, we measure the average accuracy in terms of the number of time steps the model has last seen the label of a particular class. It is reported in Table 4 and in Appendix C Table 13, 14, where we directly compare CPM and OPN to see the effect of temporal context. CPM is significantly better than OPN on 1-shot within a short interval, which suggests that the contextual RNN makes the recall of the recent past much easier. On RoamingImageNet, OPN eventually surpasses CPM on longer horizon, and this can be explained by the fact that OPN has more stable prototypes, whereas prototypes in CPM could potentially be affected by the fluctuation of the contextual RNN over a longer horizon.

**Ablation studies:** We ablate each individual module we introduce. Results are shown in Tables 5 and 6. Table 5 studies different ways we use the RNN, including the context vector $h^{\text{RNN}}$, the predicted threshold parameters $\beta_t^*, \gamma_t^*$, and the predicted metric scaling vector $_t$. Table 6 studies various ways to learn from unlabeled examples, where we separately disable the RNN update, prototype update, and distinct write-threshold parameters $\beta_t^w, \gamma_t^w$ (vs. using read-threshold parameters), which makes it robust to potential mistakes made in semi-supervised learning. We verify that each component has a positive impact on the performance.

## 6 CONCLUSION

We proposed online contextualized few-shot learning, OC-FSL, a paradigm for machine learning that emulates a human or artificial agent interacting with a physical world. It combines multiple properties to create a challenging learning task: every input must be classified or flagged as novel, every input is also used for training, semi-supervised learning can potentially improve performance, and the temporal distribution of inputs is non-IID and comes from a generative model in which input and class distributions are conditional on a latent environment with Markovian transition probabilities. We proposed the RoamingRooms dataset to simulate an agent wandering within a physical world. We also proposed a new model, CPM, which uses an RNN to extract spatiotemporal context from the input stream and to provide control settings to a prototype-based FSL model. In the context of naturalistic domains like RoamingRooms, CPM is able to leverage contextual information to attain performance unmatched by other state-of-the-art FSL methods.

**Acknowledgments:** We thank Fei Sha, James Lucas, Eleni Triantafillou and Tyler Scott for helpful feedback on an earlier draft of the manuscript. Resources used in preparing this research were provided, in part, by the Province of Ontario, the Government of Canada through CIFAR, and companies sponsoring the Vector Institute (www.vectorinstitute.ai/#partners). This project is supported by NSERC and the Intelligence Advanced Research Projects Activity (IARPA) via Department of Interior/Interior Business Center (DoI/IBC) contract number D16PC00003. The U.S. Government is authorized to reproduce and distribute reprints for Governmental purposes notwithstanding any copyright annotation thereon. Disclaimer: The views and conclusions contained herein are those of the authors and should not be interpreted as necessarily representing the official policies or endorsements, either expressed or implied, of IARPA, DoI/IBC, or the U.S. Government.

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

# A  DATASET DETAILS

## A.1  BENCHMARK COMPARISON

We include Table 8 to compare existing continual and few-shot learning paradigms.

## A.2  ROAMINGOMNIGLOT & ROAMINGIMAGENET SAMPLER DETAILS

For the RoamingOmniglot and the RoamingImageNet experiments, we use sequences with maximum 150 images, from 5 environments. For individual environment, we use a Chinese restaurant process to sample the class distribution. In particular, the probability of sampling a new class is:

$$p_{\text{new}} = \frac{k\alpha + \theta}{m + \theta},\tag{10}$$

where $k$ is the number of classes that we have already sampled in the environment, and $m$ is the total number of instances we have in the environment. $\alpha$ is set to 0.2 and $\theta$ is set to 1.0 in all experiments.

The environment switching is implemented by a Markov switching process. At each step in the sequence there is a constant probability $p_{\text{switch}}$ that switches to another environment. For all experiments, we set $p_{\text{switch}}$ to 0.2. We truncate the maximum number of appearances per class to 6. If the maximum appearance is reached, we will sample another class.

## A.3  METRICS

**Average precision:**  We chose to use AP (average precision or area under the precision-recall curve) as a way of integrating two aspects of performance:

1. the binary accuracy of whether an instance belongs to a known or unknown class (KU-Assign for short), and
2. the accuracy of assigning an instance the correct class label given it is from a known class (Class-Assign for short).

The procedure to calculate AP is as follows. We first sort all the KU-Assign, Class-Assign predictions across all sequences in descending order based on KU-Assign probability, where the high ranked predictions should be known (not novel) classes. For the N top ranked instances in the sorted list, we compute:

1. precision@N = correct(Class-Assign)@N / N
2. recall@N = correct(Class-Assign)@N / K,

where K is the true number of known instances and correct(Class-Assign)@N is the count of the number of correct class assignments among the top N. (The class assignment for an unknown instance is always incorrect.) To obtain the AP, we compute the integral of the function (y=precision@N, x=recall@N) across all N's.

**N-shot accuracy:**  We define N-shot accuracy as the number of times an instance that has been seen N times thus far in the sequence is classified correctly. We compute the mean and standard error of this over all sequences.

## A.4  ADDITIONAL ROAMINGROOMS STATISTICS

Statistics of the RoamingRooms are included in Table 7, in comparison to other few-shot and continual learning datasets. Note that since RoamingRooms is collected from a simulated environment, with 90 indoor worlds consisting of 1.2K panorama images and 1.22M video frames. The dataset contains about 6.9K random walk sequences with a maximum of 200 frames per sequence. For training we randomly crop 100 frames to form a training sequence. There are 7.0K unique instance classes.

Plots of additional statistics of RoamingRooms are shown in Figure 7. In addition to the ones shown in the main paper, instances and viewpoints also follow long tail distributions. The number of objects in each frame follows an exponential distribution.

Table 7: Continual & few-shot learning datasets

|  | Images | Sequences | Classes | Content |
|---|---|---|---|---|
| Permuted MNIST (Lecun et al., 1998) | 60K | - | - | Hand written digits |
| Omniglot (Lake et al., 2015) | 32.4K | - | 1.6K | Hand written characters |
| CIFAR-100 (Krizhevsky, 2009) | 50K | - | 100 | Common objects |
| mini-ImageNet (Vinyals et al., 2016) | 50K | - | 100 | Common objects |
| tiered-ImageNet (Ren et al., 2018) | 779K | - | 608 | Common objects |
| OpenLORIS (She et al., 2019) | 98K | - | 69 | Small table-top obj. |
| CORe50 (Lomonaco & Maltoni, 2017) | 164.8K | 11 | 50 | Hand-held obj. |
| RoamingRooms (Ours) | 1.22M | 6.9K | 7.0K | General indoor instances |

## A.5 ROAMINGROOMS SIMULATOR DETAILS

We generate our episodes with a two-stage process using two simulators – HabitatSim (Savva et al., 2019) and MatterSim (Anderson et al., 2018) – because HabitatSim is based on 3D meshes and using HabitatSim alone will result in poor image quality due to incorrect mesh reconstruction. Therefore we sacrificed the continuous movement of agents within HabitatSim and base our environment navigation on the discrete viewpoints in MatterSim, which is based on real panoramic images. The horizontal field of view is set to 90 degrees for HabitatSim and 100 degrees for MatterSim, and we simulate with $800 \times 600$ resolution.

The first stage of generation involves randomly picking a sequence of viewpoints on the connectivity graph within MatterSim. For each viewpoint, the agent scans the environment along the yaw and pitch axes for a random period of time until a navigable viewpoint is within view. The time spent in a single viewpoint follows a Gaussian distribution with mean 5.0 and standard deviation 1.0. At the start of each new viewpoint, the agent randomly picks a direction to rotate and takes 12.5 degree steps along the yaw axis, and with 95% probability, a 5 degree rotation along the pitch axis is applied in a randomly chosen direction. When a navigable viewpoint is detected, the agent will navigate to the new viewpoint and reset the direction of scan. When multiple navigable viewpoints are present, the agent uniformly samples one.

In the second stage, an agent in HabitatSim retraces the viewpoint path and movements of the first stage generated by MatterSim, collecting mesh-rendered RGB and instance segmentation sensor data. The MatterSim RGB and HabitatSim RGB images are then aligned via FLANN-based feature matching (Muja & Lowe, 2009), resulting in an alignment matrix that is used to place the MatterSim RGB and HabitatSim instance segmentation maps into alignment. The sequence of these MatterSim RGB and HabitatSim instance segmentation maps constitute an episode.

We keep objects of the following categories: `picture`, `chair`, `lighting`, `cushion`, `table`, `plant`, `chest of drawers`, `towel`, `sofa`, `bed`, `appliances`, `stool`, `tv monitor`, `clothes`, `toilet`, `fireplace`, `furniture`, `bathtub`, `gym equipment`, `blinds`, `board panel`. We initially generate 600 frames per sequence and remove all the frames with no object. Then we store every 200 image frames into a separate file.

During training and evaluation, each video sequence is loaded, and for each image we go through each object present in the image. We create the attention map using the segmentation groundtruth of the selected object. The attention map and the image together form a *frame* in our model input. For training, we randomly crop 100 frames from the sequence, and for evaluation we use the first 100 frames for deterministic results.

Please visit our released code repository to download the RoamingRooms dataset.

## A.6 SEMI-SUPERVISED LABELS:

Here we describe how we sample the labeled vs. unlabeled flag for each example in the semi-supervised sequences in both RoamingOmniglot and RoamingRooms datasets. Due to the imbalance in our class distribution (from both the Chinese restaurant process and real data collection), directly masking the label may bias the model to ignore the rare seen classes. Ideally, we would like to preserve at least one labeled example for each class. Therefore, we designed the following procedure.

Table 8: Comparison of past FSL and CL paradigms vs. our online contextualized FSL (OC-FSL).

| Tasks | Few Shot | Semi-sup. Supp. Set | Continual | Online Eval. | Predict New | Soft Context Switch |
|---|---|---|---|---|---|---|
| Incremental Learning (IL) (Rebuffi et al., 2017) | ○ | ○ | ● | ◑ | ○ | ○ |
| Few-shot (FSL) (Vinyals et al., 2016) | ● | ○ | ○ | ○ | ○ | ○ |
| Incremental FSL (Ren et al., 2019) | ● | ○ | ◑ | ○ | ○ | ○ |
| Cls. Incremental FSL (Tao et al., 2020) | ● | ○ | ● | ◑ | ○ | ○ |
| Semi-supv. FSL (Ren et al., 2018) | ● | ● | ○ | ○ | ● | ○ |
| MOCA (Harrison et al., 2019) | ● | ○ | ● | ○ | ○ | ◑ |
| Online Mixture (Jerfel et al., 2019) | ● | ○ | ● | ○ | ○ | ◑ |
| Online Meta (Javed & White, 2019) | ● | ○ | ● | ○ | ○ | ○ |
| Continual FSL* (Antoniou et al., 2020) | ● | ○ | ● | ○ | ○ | ○ |
| OSAKA* (Caccia et al., 2020) | ● | ○ | ● | ● | ◑ | ● |
| OC-FSL (Ours) | ● | ● | ● | ● | ● | ● |

\* denotes concurrent work.

First, for each class $k$, suppose $m_k$ is the number of examples in the sequence that belong to the class. Let $\alpha$ be the target label ratio. Then the class-specific label ratio $\alpha_k$ is:

$$\alpha_k = (1 - \alpha) \exp(-0.5(m_k - 1)) + \alpha. \tag{11}$$

We then for each class $k$, we sample a binary Bernoulli sequence based on $\mathrm{Ber}(\alpha_k)$. If a class has all zeros in the Bernoulli sequence, we flip the flag of one of the instances to 1 to make sure there is at least one labeled instance for each class. For all experiments, we set $\alpha = 0.3$.

## A.7 DATASET SPLITS

We include details about our dataset splits in Table 9 and 10.

## B EXPERIMENT DETAILS

### B.1 NETWORK ARCHITECTURE

For the RoamingOmniglot experiment we used the common 4-layer CNN for few-shot learning with 64 channels in each layer, resulting in a 64-d feature vector (Snell et al., 2017). For the RoamingRooms experiment we resize the input to 120×160 and we use the ResNet-12 architecture (Oreshkin et al., 2018) with {32,64,128,256} channels per block. To represent the feature of the input image with an attention mask, we concatenate the global average pooled feature with the attention ROI feature, resulting in a 512d feature vector. For the contextual RNN, in both experiments we used an LSTM (Hochreiter & Schmidhuber, 1997) with a 256d hidden state.

We use a linear layer to map from the output of the RNN to the features and control variables. We obtain $\gamma^{r,w}$ by adding 1.0 to the linear layer output and then applying the softplus activation. The bias units for $\beta^{r,w}$ are initialized to 10.0. We also apply the softplus activation to  from the linear layer output.

### B.2 TRAINING PROCEDURE

We use the Adam optimizer (Kingma & Ba, 2015) for all of our experiments, with a gradient cap of 5.0. For RoamingOmniglot we train the network for 40k steps with a batch size 32 and maximum sequence length 150 across 2 GPUs and an initial learning rate 2e-3 decayed by 0.1× at 20k and 30k steps. For RoamingRooms we train for 20k steps with a batch size 8 and maximum sequence length 100 across 4 GPUs and an initial learning rate 1e-3 decayed by 0.1× at 8k and 16k steps. We use the BCE coefficient $\lambda = 1$ for all experiments. In semi-supervised experiments, around 30% examples are labeled when the number of examples grows large ($\alpha = 0.3$, see Equation 11). Early stopping is used in RoamingRooms experiments where the checkpoint with the highest validation AP score is chosen. For RoamingRooms, we sample Bernoulli sequences on unlabeled inputs to gradually allow semi-supervised writing to the prototype memory and we find it helps training stability. The probability starts with 0.0 and increase by 0.2 every 2000 training steps until reaching 1.0.

Table 9: **Split information for** *RoamingOmniglot*. Each column is an alphabet and we include all the characters in the alphabet in the split. Rows are continuation of lines.

| | | | |
|---|---|---|---|
| Train | Angelic
Aurek-Besh
Asomtavruli
Korean
Alphabet of the Magi
Tagalog
Cyrillic
Gujarati
Atlantean
Atemayar Qelisayer
Latin | Grantha
Japanese (hiragana)
Sanskrit
Arcadian
Blackfoot
Anglo-Saxon Futhorc
Burmese
Ge ez
Japanese (katakana)
Glagolitic
Inuktitut | N Ko
Malay
Ojibwe
Greek
Futurama
Braille
Avesta
Syriac (Estrangelo)
Balinese
Tifinagh |
| Val | Hebrew
Early Aramaic | Mkhedruli
Bengali | Armenian |
| Test | Gurmukhi
Malayalam
Old Church Slavonic
Sylheti
ULOG | Kannada
Manipuri
Oriya
Tengwar | Keble
Mongolian
Syriac (Serto)
Tibetan |

Table 10: **Split information for** *RoamingRooms*. Each column is the ID of an indoor world. Rows are continuation of the lines.

| | | | | | |
|---|---|---|---|---|---|
| Train | r1Q1Z4BcV1o
8WUmhLawc2A
gYvKGZ5eRqb
VVfe2KiqLaN
2t7WUuJeko7
1pXnuDYAj8r
TbHJrupSAjP
S9hNv5qa7GM
VzqfbhrpDEA
rqfALeAoiTq
759xd9YjKW5
GdvgFV5R1Z5 | JmbYfDe2QKZ
Uxmj2M2itWa
gxdoqLR6rwA
fzynW3qQPVF
pLe4wQe7qrG
b8cTxDM8gDG
sKLMLpTHeUy
EDJbREhghzL
D7G3Y4RVNrH
e9zR4mvMWw7
wc2JMjhGNzB
kEZ7cmS4wCh | 29hnd4uzFmX
mJXqzFtmKg4
YFuZgdQ5vWj
WYY7iVyf5p8
cV4RVeZvu5T
x8F5xyUWy9e
2azQ1b91cZZ
qoiz87JEwZ2
ZMojNkEp431
yqstnuAEVhm
rPc6DW4iMge
vyrNrziPKCB | ULsKaCPVFJR
V2XKFyX4ASd
gTV8FGcVJC9
VFuaQ6m2Qom
XcA2TqTSSAj
X7HyMhZNoso
2n8kARJN3HM
q9vSo1VnCiC
uNb9QFRL6hY
zsNo4HB9uLZ
jh4fc5c5qoQ
D7N2EKCX4Sj | E9uDoFAP3SH
EU6Fwq7SyZv
sT4fr6TAbpF
YmJkqBEsHnH
ur6pFq6Qu1A
aayBHfsNo7d
Vvot9Ly1tCj
Vt2qJdWjCF2
5LpN3gDmAk7
JF19kD82Mey
HxpKQynjfin
PX4nDJXEHrG |
| Val | s8pcmisQ38h
jtcxE69GiFV | dhjEzFoUFzH
QUCTc6BB5sX | RPmz2sHmrrY
p5wJjkQkbXX | 1LXtFkjw3qL
JeFG25nYj2p | 8194nk5LbLH
82sE5b5pLXE |
| Test | oLBMNvg9in8
SN83YJsR3w2
UwV83HsGsw3
PuKPg4mmafe | r47D5H71a5s
gZ6f7yhEvPG
VLzqgDo317F
Pm6F8kyY3z2 | Z6MFQCViBuw
ac26ZMwG7aT
17DRP5sb8fy
i5noydFURQK | YVUC4YcDtcY
7y3sRwLe3Va
pa4otMbVnkk
ARNzJeq3xxb | pRbA3pwrgk9
B6ByNegPMKs
5ZKStnWn8Zo
5q7pvUzZiYa |

## B.3 DATA AUGMENTATION

For RoamingOmniglot, we pad the 28×28 image to 32×32 and then apply random cropping.

For RoamingRooms, we apply random cropping in the time dimension to get a chunk of 100 frames per input example. We also apply random dropping of 5% of the frames. We pad the 120×160 images to 126 × 168 and apply random cropping in each image frame. We also randomly flip the order of the sequence (going forward or backward).

## B.4 SPATIOTEMPORAL CONTEXT EXPERIMENT DETAILS

We use the Kylberg texture dataset (Kylberg, 2011) without rotations. Texture classes are split into train, val, and test, defined in Table 12. We resize all images first to 256×256. For each Omniglot image, a 28×28 patch is randomly cropped from a texture image to serve as background. Random Gaussian noises with mean zero and standard deviation 0.1 are added to the background images.

Table 11: *RoamingRooms* dataset split size

| Split | Worlds | Sequences | Frames |
|-------|--------|-----------|--------|
| Train | 60 | 4,699 | 823,444 |
| Val | 20 | 725 | 125,823 |
| Test | 10 | 1,547 | 271,335 |
| Total | 90 | 6,971 | 1,220,602 |

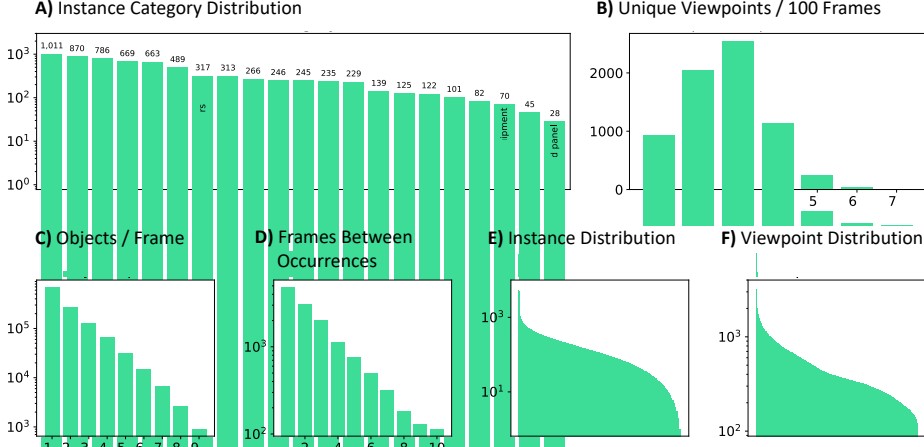

Figure 7: Additional statistics about our RoamingRooms dataset.

For spatial background experiments, we added an additional learnable network of the same size as the main network to take the background image as input, and output the same sized embedding vector. This embedding vector is further concatenated with the main embedding vector to form the final embedding of the input. We also found that using spatially overlayed images with a single CNN can achieve similar performance as well. The final numbers are reported using the concatenation approach since it is less prone to overlay noises and is more similar to the implementation we use in the RoamingRooms experiments.

### B.5 BASELINE IMPLEMENTATION DETAILS

**Online meta-learning (OML):** The OML model performs one gradient descent step for each input. In order for the model to predict unknown, we use the probability output from the softmax layer summing across the unused units. For example, if the softmax layer has 40 units and we have only seen 5 classes so far, then we sum the probability from the 6th to the last units. This summed probability is separately trained with a binary cross entropy, same as in Equation 9.

The inner learning rate is set to 1e-2 and we truncate the number of unrolled gradient descent steps to 5/20 (RoamingOmniglot/RoamingRooms), in order to make the computation feasible. For RoamingOmniglot, the network is trained with a batch size 32 across 2 GPUs, for a total of 20k steps, with an initial learning rate 2e-3 decayed by 0.1 at 10k and 16.7k steps. For RoamingRooms, the network is trained with a batch size 8 across 4 GPUs, for a total of 16k steps, with an initial learning rate 1e-3 decayed by 0.1 at 6.4k and 12.8k steps.

**Long short-term memory (LSTM):** We apply a stacked two layer LSTM with 256 hidden dimensions. Inputs are $\mathbf{h}_t^{\text{CNN}}$ concatenated with the label one-hot vector. If an example is unlabeled, then the label vector is all-zero. We directly apply a linear layer on top of the LSTM to map the LSTM memory output into classification logits, and the last logit is the binary classification logit reserved for unknown. The training procedure is the same as our CPM model.

**Differentiable neural computer (DNC):** In order to make the DNC model work properly, we found that it is sometimes helpful to pretrain the CNN weights. Simply initializing from scratch and train CNN+DNC end-to-end sometimes results in poor performance. We hypothesize that the attention structure in the DNC model is detrimental to representation learning. Therefore,

Table 12: **Split information for the Kylberg texture dataset**. Each column is an texture type. Rows are continuation of lines.

|       |              |          |           |          |           |
|-------|--------------|----------|-----------|----------|-----------|
| Train | blanket2     | ceiling2 | floor2    | grass1   | linseeds1 |
|       | pearlsugar1  | rice2    | scarf2    | screen1  | seat2     |
|       | sesameseeds1 | stone1   | stoneslab1|          |           |
| Val   | blanket1     | canvas1  | ceiling1  | floor1   | scarf1    |
|       | rice1        | stone2   |           |          |           |
| Test  | wall1        | lentils1 | cushion1  | rug1     | sand1     |
|       | oatmeal1     | stone3   | seat1     |          |           |

for RoamingOmniglot experiments, we use pretrained ProtoNet weights for solving 1-shot 5-way episodes to initialize the CNN, and we keep finetuning the CNN weights with 10% of the full learning rate. For RoamingRooms experiments, we train the full model end-to-end from scratch.

The DNC is also modified so that it is more effective using the label information from the input. In the original MANN paper (Santoro et al., 2016) for one-shot learning, the input features $\mathbf{h}_t^{\text{CNN}}$ and the label one-hot ID are simply concatenated to feed into the LSTM controller of MANN. We find that it is beneficial to directly add label one-hot vector as an input to the write head that generates the write attention and the write content. Similar to the LSTM model, the memory readout is also sent to a linear layer in order to get the final classification logits, and the last logit is the binary classification logit reserved for the unknowns. Finally we remove the linkage prediction part of the DNC due to training instability.

The controller LSTM has 256 hidden dimensions, and the memory has 64 slots each with 64 dimensions. There are 4 read heads and 4 write heads. The training procedure is the same as CPM.

**Online ProtoNet:**   Online ProtoNet is our modification of the original ProtoNet (Snell et al., 2017). It is similar to our CPM model without the contextual RNN. The feature from the CNN is directly written to the prototype memory. In addition, we do not predict the control hyperparameters $\beta_t^{\{r,w\}}, \gamma_t^{\{r,w\}}$ from the RNN and they are learned as regular parameters. The training procedure is the same as CPM.

**Online MatchingNet:**   Online MatchingNet is our modification of the original Matching-Net (Vinyals et al., 2016). We do not consider the context embedding in the MatchingNet paper since it was originally designed for the entire episode using an attentional RNN encoder. It is similar to online ProtoNet but instead of doing online averaging, it directly stores each example and its class. Since it is an example-based storage, we did not extend it to learn from unlabeled examples, and all unlabeled examples are skipped. We use a similar decision rule to determine whether an example belongs to a known cluster by looking at the distance to the nearest exemplar stored in the memory, shifted by $\beta$ and scaled by $1/\gamma$. Note that online MatchingNet is not efficient at memory storage since it scales with the number of steps in the sequence. In addition, we use the negative Euclidean distance as the similarity function. The training procedure is the same as CPM.

**Online infinite mixture prototypes (IMP):**   Online IMP is proposed as a mix of prototype and example-based storage by allowing a class to have multiple clusters. If an example is classified as unknown or it is unlabeled, we will assign its cluster based on our prediction, which either assigns it to one of the existing clusters or creates a new cluster, depending on its distance to the nearest cluster. If a cluster with an unknown label later is assigned with an example with a known class, then the cluster label will also be updated. We use the same decision rule as online ProtoNet to determine whether an example belongs to a known cluster by looking at the distance to the nearest cluster, shifted by $\beta$ and scaled by $1/\gamma$. As described above, online IMP has the capability of learning from unlabeled examples, unlike online MatchingNet. However similar to online MatchingNet, online IMP is also not efficient at memory storage since in the worst case it also scales with the number of steps in the sequence. Again, the training procedure is the same as CPM.

Table 13: **Effect of forgetting over a time interval on RoamingRooms.** Average accuracy vs. the number of time steps since the model has last seen the label of a particular class.

| | Supervised | | | | | | Semi-Supervised | | | | | |
| --- | --- | --- | --- | --- | --- | --- | --- | --- | --- | --- | --- | --- |
| | 1 - 2 | 3 - 5 | 6 - 10 | 11 - 20 | 21 - 50 | 51 - 100 | 1 - 2 | 3 - 5 | 6 - 10 | 11 - 20 | 21 - 50 | 51 - 100 |
| OPN 1-Shot | 93.5 | 89.3 | 79.4 | 67.2 | 60.3 | 60.1 | 86.5 | 83.6 | 76.3 | 68.4 | 64.7 | 61.5 |
| CPM 1-Shot | **95.7** | **92.2** | **85.7** | **75.2** | **70.0** | **66.4** | **91.0** | **88.7** | **82.9** | **77.0** | **72.2** | **66.5** |
| OPN 3-Shot | 95.1 | 91.8 | 85.6 | 78.2 | 74.6 | 73.8 | 92.6 | 88.0 | 85.1 | 81.1 | 80.6 | 76.7 |
| CPM 3-Shot | **96.1** | **93.8** | **87.7** | **81.4** | **79.1** | **78.2** | **94.8** | **91.0** | **86.9** | **83.1** | **82.7** | **79.2** |

Table 14: **Effect of forgetting over a time interval on RoamingImageNet.** Average accuracy vs. the number of time steps since the model has last seen the label of a particular class.

| | Supervised | | | | | | Semi-Supervised | | | | | |
| --- | --- | --- | --- | --- | --- | --- | --- | --- | --- | --- | --- | --- |
| | 1 - 2 | 3 - 5 | 6 - 10 | 11 - 20 | 21 - 50 | 51 - 100 | 1 - 2 | 3 - 5 | 6 - 10 | 11 - 20 | 21 - 50 | 51 - 100 |
| OPN 1-Shot | 40.8 | 35.9 | 33.0 | **30.7** | **27.0** | **21.4** | 40.5 | 37.7 | 35.9 | **33.7** | **31.5** | **28.4** |
| CPM 1-Shot | **67.5** | **52.9** | **35.5** | 24.2 | 18.3 | 13.8 | **60.4** | **51.3** | **39.5** | 26.6 | 21.8 | 15.4 |
| OPN 3-Shot | 52.5 | 50.3 | **48.8** | **47.2** | **44.4** | **42.3** | 57.6 | 55.1 | **54.6** | **52.3** | **52.1** | **49.5** |
| CPM 3-Shot | **77.8** | **64.5** | 46.6 | 32.9 | 24.7 | 17.9 | **76.1** | **61.8** | 48.6 | 30.5 | 24.1 | 15.6 |

# C   ADDITIONAL EXPERIMENTAL RESULTS

## C.1   EFFECT OF FORGETTING

We report the effect of forgetting of RoamingRooms and RoamingImageNet in Table 13 and 14.

## C.2   EMBEDDING VISUALIZATION

Figure 8 shows the learned embedding of each example in Online ProtoNet vs. our CPM model in RoamingOmniglot sequences, where colors indicate environment IDs. In Online ProtoNet, the example features does not reflect the temporal context, and as a result, colors are scattered across the space. By contrast, in the CPM embedding visualization, colors are clustered together and we see a smoother transition of environments in the embedding space.

## C.3   CONTROL PARAMETERS VS. TIME

Finally we visualize the control parameter values predicted by the RNN in Figure 9. We verify that we indeed need two sets of $\beta$ and $\gamma$ for read and write operations separately as they learn different values. $\beta^w$ is smaller than $\beta^r$ which means that the network is more conservative when writing to prototypes. $\gamma^w$ grows larger over time, which means that the network prefers a softer slope when writing to prototypes since in the later stage the prototype memory has already stored enough content and it can grow faster, whereas in the earlier stage, the prototype memory is more conservative to avoid embedding vectors to be assigned to wrong clusters.

Online ProtoNet          CPM (Ours)

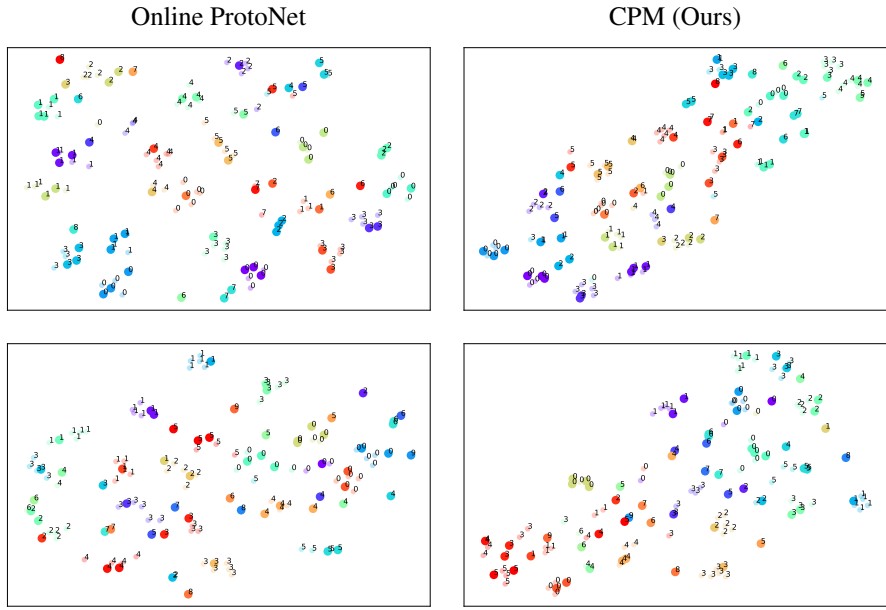

Figure 8: **Embedding space visualization of RoamingOmniglot sequences using t-SNE (Maaten & Hinton, 2008)**. Different color denotes different environments. Text labels (relative to each environment) are annotated beside the scatter points. Unlabeled examples shown in smaller circles with lighter colors. **Left:** Online ProtoNet; **Right:** CPM. The embeddings learned CPM model shows a smoother transition of classes based on their temporal environments.

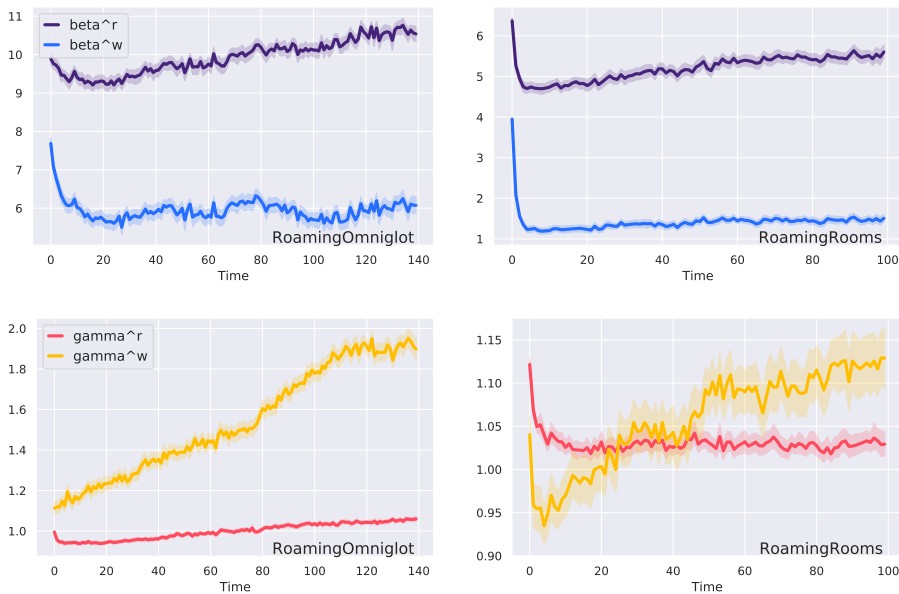

Figure 9: **CPM control parameters ($\beta^{r,w}, \gamma^{r,w}$) vs. time. Left:** RoamingOmniglot sequences; **Right:** RoamingRooms sequences; **Top:** $\beta^{r,w}$ the threshold parameter; **Bottom:** $\gamma^{r,w}$ the temperature parameter.

