# OpenReview forum: "Wandering within a world: Online contextualized few-shot learning"
_ICLR.cc/2021/Conference — ICLR 2021 Poster_

### Official Review · AnonReviewer2 · 2020-10-28
**Interesting continual learning paradigm with few shot learning**

**Rating:** 7
**Confidence:** 3

**Review:**

The paper proposes a new learning paradigm that combines both few-shot learning(FSL)  and continual learning (CL) to provide a more realistic learning environment rather than the traditional train-test-retrain approach in FSL. Two environments are proposed, along with a novel dataset. The evaluation seems to be thorough, with strong baselines (conventional approaches adapted to the proposed setting). A novel approach is proposed based on augmenting ProtoNets with contextual memory and is shown to have consistently strong performance compared to the baselines on both tasks.

Strengths:
+ The paper is very well written and reads very nicely. I particularly liked the motivation for the task.
+ The use of contextual memory (incorporating both spatial and temporal context) is very interesting and is a promising approach for both FSL as well as a general learning architecture for visual event perception.
Concerns:
- While very excited by the potential of the proposed learning environment, I am a bit confused about how the actual implementation/evaluation maps to the motivation. For example, from what I can see, all baselines (including the proposed model) have access to the number of classes that are present in the data (k). A softmax-based decision function forces the model to choose one of these classes, either based on some online learning-based features or just through storing examples. Now, the premise (that of knowing when and what to learn) is not quite satisfied here and thus the metric (Average Precision) doesn't quite capture the entire picture. I think a better metric, IMHO, would be the harmonic mean of novel class detection and known class prediction. This would allow us to actually ascertain whether the models are learning novel instances and not just getting "lucky".
- The baselines that are chosen are classic approaches to few-shot learning. Not many continual learning approaches are tested and I think that would make for a better comparison. Again, the actual learning setting is quite a bit more simplified than what is claimed.

Overall, I think it is a very nicely written paper with some issues with evaluation settings that might be exaggerating the performance of baselines.

==== Post discussion Update ====

I am updating my score to accept after the discussion.

---

> ### Author Response · Authors · 2020-11-21
> **Response to Reviewer 2**
>
> Thank you for your insightful comments. Below are our responses:
>
> > I am a bit confused about how the actual implementation/evaluation maps to the motivation. For example, from what I can see, all baselines (including the proposed model) have access to the number of classes that are present in the data (k).
>
> All models have access to the number of classes *that have been learned so far up until the current step*, but they are not given the total number of classes that will appear in the sequence.
>
> > A softmax-based decision function forces the model to choose one of these classes, either based on some online learning-based features or just through storing examples. Now, the premise (that of knowing when and what to learn) is not quite satisfied here and thus the metric (Average Precision) doesn't quite capture the entire picture. I think a better metric, IMHO, would be the harmonic mean of novel class detection and known class prediction. This would allow us to actually ascertain whether the models are learning novel instances and not just getting "lucky".
>
> To clarify, our model has two-headed outputs. It separately outputs 1) a sigmoid probability of whether it belongs to a new class, and 2) a softmax probability of which previously known class it belongs to.
>
> We chose to use AP (average precision or area under the precision-recall curve) as a way of integrating two aspects of performance:
>
> 1) the binary accuracy of whether an instance belongs to a known or unknown class (KU-Assign for short), and
>
> 2) the accuracy of assigning an instance the correct class label given it is from a known class (Class-Assign for short).
>
> The procedure to calculate AP is as follows. We first sort all the {KU-Assign, Class-Assign} predictions across all sequences in descending order based on KU-Assign probability, where the high ranked predictions should be known (not novel) items. For the N top ranked items in the sorted list, we compute:
>
> * precision@N = correct(Class-Assign)@N / N
> * recall@N = correct(Class-Assign)@N / K,
>
> where K is the true number of known instances and correct(Class-Assign)@N is the count of the number of correct class assignments among the top N. (The class assignment for an unknown item is always incorrect.) To obtain the AP, we compute the integral of the function (y=precision@N, x=recall@N) across all N’s.
>
> Therefore, just like you suggested, it is a combination of novel class detection and known class prediction: precision takes in the accuracy of known class prediction, and for novel class detection, you need to have a perfect sigmoid score ranking. A harmonic mean will also show similar trends, but would be a single point on the precision-recall curve.
>
> > The baselines that are chosen are classic approaches to few-shot learning. Not many continual learning approaches are tested and I think that would make for a better comparison. Again, the actual learning setting is quite a bit more simplified than what is claimed.
>
> This is a good point. We consider our setup primarily an extension of few-shot learning, so we feel it is the most appropriate to compare with few-shot learning baselines. However it is somewhere between few-shot and continual learning. So we do in fact compare to both. OML was recently proposed as a meta-continual learning model. And we have adapted it to output unknown classes (OML-U and OML-U++) and compared them. While we would love to compare to more continual learning methods, we need to do some non-trivial amount of adaptation to each, to have a novelty detection output. The length of each online sequence in our setting is also shorter than most continual learning setups.

---

> > ### Comment · AnonReviewer2 · 2020-11-24
> > **Thanks for the clarification**
> >
> > Thanks for providing the clarifications. I really appreciate your detailed responses. Having read the other reviews and the author's responses, I feel that the paper makes a good contribution by introducing a novel task setup which is challenging and is more realistic. Overall, I think this is a good paper and would recommend its acceptance.

---

### Official Review · AnonReviewer1 · 2020-10-28
**A novel problem formulation with a more realistic evaluation setting, a novel data set, and a new method that works well in this setting**

**Rating:** 7
**Confidence:** 4

**Review:**

############## Summary ##############

This paper presents a new definition of the continual learning problem which attempts to bridge the gap between artificial settings typically used to evaluate continual learning and the way the real world requires us humans to perform. Concretely, this setting involves changing environments, where the agent is expected to autonomously detect when new classes are encountered. The changes in the environment include both spatial and temporal cues to enable the agent to differentiate between environments. Additionally, this work creates a new data set and adapts an existing one to this novel setting, and provides a method that can deal with it, as validated empirically.

############## Strengths ##############

1. The provided evaluation setting is close to what we would want in a continual learning setting.
2. The provided data sets are potentially useful for evaluating future algorithms.
3. The presented method is effective in handling this challenging evaluation setting.

############## Weaknesses ##############

1. While the evaluation setting is closer to a human setting, it appears that there is a pre-training stage where the agent prepares for that evaluation setting, which is not described in detail.
2. The techniques used to create the Roaming version of Omniglot could potentially be applied to different existing data sets, but this is not described in the paper, limiting the usefulness of it.
3. Since the authors do not create additional benchmarks following this procedure, the evaluation is only on two data sets, which limits the reader's ability to assess the benefits of the proposed approach.

############## Recommendation ##############

I recommend this paper for acceptance. I believe that problem settings that become increasingly realistic is a valuable direction of work, and although this manuscript perhaps oversells how realistic their proposed setting is, I believe it is still a step in the right direction. The introduction of a novel data set on Matterport3D could be impactful to future researchers working in this field, and provides a more realistic benchmark than many existing ones. Finally, the proposed approach is novel and effective.


############## Arguments ##############

The main contribution of this work is to propose a novel problem formulation for continual few-shot learning, which is closer to realistic continual learning. I believe this in itself is interesting. The evaluation setting requires the agent to automatically detect when new classes are encountered, and simultaneously classify objects into existing classes. However, from the experimental section, it appears that the authors' proposed method requires a pre-training phase, where the agent encounters many evaluation scenarios like this one, in order to perform well in the final evaluation scenario. This is never explicitly stated or explained in detail, but if it is the case, it is highly unrealistic, as it requires the agent to essentially live multiple lives, before being able to perform well in a "test" life. I would encourage the authors to explain this in more detail, and more honestly assess how realistic their problem formulation is. Even if the pre-training phase is unrealistic, the evaluation phase is still realistic, which is valuable. This should be explicitly stated to make the contributions of the paper clearer.

I very much liked the proposed RoamingRooms data set, and I also believe that to be a valuable contribution. The fact that a similar data set can be created from Omniglot interesting. Would it be possible to create "Roaming" versions of other synthetic benchmarks? What are the limitations for this? Since we typically expect continual learning methods to be evaluated in various benchmarks, it would be valuable to see diverse benchmarks for this new setting, or at least the details for how to create them. This could also enable a more comprehensive evaluation of the proposed approach in this submission.

The idea of using the recurrent net to output the thresholds used to make decisions of when to detect instances as novel is quite interesting, but I was left lacking an intuitive description of what this should do and how.

In terms of the empirical results, I believe them to demonstrate a substantial margin of improvement with respect to baselines, and the set of baselines to be sufficient. The ablative tests show the usefulness of each part of the proposed approach.

############## Additional feedback ##############

The following points are provided as feedback to hopefully help better shape the submitted manuscript, but did not impact my recommendation in a major way.

Intro
- What is a trial?
- Hyperref to Figure 4 leads to Section 4.
- This section states that comparisons will be made against few-shot baslines, which made me think that no comparisons would be made against continual learning baselines. Please clarify that you indeed compare against continual learning baselines.

Sec 2
- Last paragraph of FSL has an incomplete sentence, and it remains unclear how Caccia et al. (2020) compares to the submission. Since this is clearly the most closely related work (as evidenced in Table 6), it is very important to make this comparison as complete as possible.
- Omniglotor --> Omniglot or
- Very comprehensive and well-written related work section, clearly outlining various lines of related work, including recent (and even concurrent) efforts that handle very similar settings to this one.

Sec 3
- Figure hyperrefs point to captions (below the figure).
- At this point, the problem formulation should clearly state how training will be carried out. This section does not mention the fact that the agent will first encounter various evaluation sequences and use those to generalize to unseen evaluation sequences. It mentions multiple few-shot sequences and train/val/test splits, but it is largely unclear at this point how these various sequences are supposed to be used.
- The way the problem setting is presented, it makes it hard to differentiate it substantially from few-shot or continual learning, since it mainly focus on evaluation being done continually. It seems like a crucial part of the distinction is that the agent is evaluated even on unseen classes.
    - This seems to be closely connected to work on open world and open set learning. Could the authors provide some guidance into how it compares to that setting? [1]

Sec 4
- When doing online average, the method seems to assume that the current prototype p_{t-1} is fixed. But, if the model parameters change over time, which I expect they do in a continual learning setting, this prototype would be different if recomputed. How is this taken this into account?

Sec 5
- I would've liked to see the ablation tests on RoamingRooms as well.
- My only concern is that the evaluations are only on two different data sets.
- In terms of the baselines, it is unclear which can leverage previous sequences to learn the new ones. I believe this is only possible with OML and variants, whereas all the other methods can only leverage data from the current sequence.


Appendices
- Thanks for providing lots of details on the experimental setting for baselines.
- The additional visualizations are useful, especially the one of the learned thresholds over time (Figure 9).
- These appendices should be referred to in the main text so the reader knows to look for them. Same for the provided videos.


[1] Geng, C., Huang, S. J., & Chen, S. (2020). Recent advances in open set recognition: A survey. IEEE Transactions on Pattern Analysis and Machine Intelligence.

---

> ### Author Response · Authors · 2020-11-21
> **Response to Reviewer 1 (Part 1)**
>
> We are grateful for R1’s detailed reading and constructive comments. Below are our responses.
>
> > It appears that the authors' proposed method requires a pre-training phase, where the agent encounters many evaluation scenarios like this one, in order to perform well in the final evaluation scenario. This is never explicitly stated or explained in detail, but if it is the case, it is highly unrealistic, as it requires the agent to essentially live multiple lives, before being able to perform well in a "test" life.
>
> We are not modeling a de-novo learner, but rather, our model is more like an adolescent who comes to new environments with some previous experience (which are encoded in the long-term memory of the CNN and RNN weights). After each episode in meta-training, the short-term memory is reset but long-term memory in the form of the CNN and RNN weights are updated. So if our work is 'lifelong learning', it's lifelong from adolescence forward. Standard FSL also has the same assumption of a fair amount of past experience prior to the first few shot-episode. Therefore, making the agent entirely starting from scratch is not our goal here.
>
> > The techniques used to create the Roaming version of Omniglot could potentially be  applied to different existing data sets, but this is not described in the paper, limiting the usefulness of it.
>
> Due to space limitations, we included the sampler details in the Appendix A.2. We have also released the code base.
>
> > Since the authors do not create additional benchmarks following this procedure, the
> > evaluation is only on two data sets, which limits the reader's ability to assess the
> > benefits of the proposed approach. Would it be possible to create "Roaming" versions
> > of other synthetic benchmarks? What are the limitations for this?
>
> We thank you for the suggestion of creating other versions such as a Roaming-ImageNet benchmark. There is no technical limitation of doing this. We would like to emphasize that our main interest is to study realistic online sequences and this is showcased in our RoamingRooms dataset, which resembles a more continuous stream of objects that could be encountered during a real-world sequence.
>
> > The idea of using the recurrent net to output the thresholds used to make decisions of
> > when to detect instances as novel is quite interesting, but I was left lacking an intuitive
> > description of what this should do and how.
>
> The recurrent net will encode a window of the previous items and use that to decide what kind of threshold should it give for known vs. unknown. For example, if the agent just entered an unfamiliar room (spatial and temporal context), then there is a higher chance that objects are new/unknown here. Also, the RNN can also control the prior belief of things being new at a given time step, since towards the end of the sequence, a greater fraction of the items are known.
>
> **Intro**
>
> > What is a trial?
>
> A trial refers to a single item in the input sequence. We will clarify the terminology.
>
> > This section states that comparisons will be made against few-shot baselines, which made me think that no comparisons would be made against continual learning baselines. Please clarify that you indeed compare against continual learning baselines.
>
> This is a good point. We consider our setup primarily an extension of few-shot learning, so we feel it is the most appropriate to compare with few-shot learning baselines. However it is somewhere between few-shot and continual learning. So we do in fact compare to both. OML was recently proposed as a meta-continual learning model. And we have adapted it to output unknown classes (OML-U and OML-U++) and compared them. While we would love to compare to more continual learning methods, we need to do some non-trivial amount of adaptation to each, to have a novelty detection output. The length of each online sequence in our setting is also shorter than most continual learning setups.
>
> **Sec 2**
>
> > Last paragraph of FSL has an incomplete sentence, and it remains unclear how Caccia et al. (2020) compares to the submission. Since this is clearly the most closely related work (as evidenced in Table 6), it is very important to make this comparison as complete as possible.
>
> Both our paper and Caccia et al. (2020) remove the notion of support and query split in each episode/sequence. However, there are still notable differences that prevent us from directly comparing them. First, their setting is not incremental class learning, and instead of detecting new classes, they detect new environments based on the current training loss. Furthermore, since in our setting, models must make  a class prediction before seeing the label; therefore it cannot use training loss to decide whether it is a new class or not. We will make Table 6 clearer w.r.t. these distinctions.

---

> > ### Author Response · Authors · 2020-11-21
> > **Response to Reviewer 1 (Part 2)**
> >
> > **Sec 3**
> >
> > > Figure hyperrefs point to captions (below the figure).
> >
> > Thanks for pointing it out. Unfortunately we don’t know how to fix this to make it point to above the figure. If you know, please let us know and we’d appreciate it!
> >
> > > At this point, the problem formulation should clearly state how training will be carried out. This section does not mention the fact that the agent will first encounter various evaluation sequences and use those to generalize to unseen evaluation sequences. It mentions multiple few-shot sequences and train/val/test splits, but it is largely unclear at this point how these various sequences are supposed to be used.
> >
> > Thanks for pointing this out. During training, models are trained with training sequences. And at test time, models are evaluated with test sequences, which contains no overlap of object classes with the training sequences. We will revise the paper and make this clearer.
> >
> > > The way the problem setting is presented, it makes it hard to differentiate it substantially from few-shot or continual learning, since it mainly focus on evaluation being done continually. It seems like a crucial part of the distinction is that the agent is evaluated even on unseen classes.
> >
> > We target this work as an extension of few-shot learning to make it an online sequential decision problem, similar to continual learning. However, in contrast to classic benchmarks in continual learning, the agent needs to quickly learn new classes with one or a few examples, and it also needs to detect unknown classes (as shown in Figure 1). We believe that being able to say something is unknown is essential for incremental class learning / continual learning but unfortunately the common definitions of continual learning currently do not capture this. We will follow your suggestion and be more explicit about detecting unknown classes.
> >
> > > This seems to be closely connected to work on open world and open set learning. Could the authors provide some guidance into how it compares to that setting? [1]
> >
> > Thank you for the reference and we will cite more open world literature in our next version. The main difference is that in our online few-shot setting, the unknown class will become known in the next immediate iteration, if the label information is given. This will require models to recognize unknown classes in a 1-shot manner.
> >
> > **Sec 4**
> >
> > > When doing online average, the method seems to assume that the current prototype p_{t-1} is fixed. But, if the model parameters change over time, which I expect they do in a continual learning setting, this prototype would be different if recomputed. How is this taken into account?
> >
> > Currently the CNN is frozen during the sequence and therefore it won’t have this issue of updating prototypes. Freezing the CNN is an approximation to the assumption that the time scale of a sequence is brief relative to the agent's lifetime. Thus, there is no representation learning within the perceptual system. However, we are currently studying the effect of changing the CNN parameters over time within the sequence.
> >
> > **Sec 5**
> >
> > > I would've liked to see the ablation tests on RoamingRooms as well.
> >
> > Thank you for the suggestion. We will run these ablation tests.
> >
> > > My only concern is that the evaluations are only on two different data sets.
> >
> > While we could run RoamingImageNet-like experiments, we would like to emphasize that our main interest is to study realistic online sequences and this is showcased in our RoamingRooms dataset.
> >
> > > In terms of the baselines, it is unclear which can leverage previous sequences to learn the new ones. I believe this is only possible with OML and variants, whereas all the other methods can only leverage data from the current sequence.
> >
> > All of the baselines leverage data only from the current sequence. During meta-learning, all models can learn the weights of the CNN and/or the RNN by rolling out the sequence and back-propagating through time.

---

> > > ### Comment · AnonReviewer1 · 2020-11-23
> > > **Updated feedback**
> > >
> > > I thank the authors for their thorough response. Some of my questions were clarified, making me more confident in my initial assessment.
> > >
> > > I believe the strengths and weaknesses I brought up initially still hold, so I retain my score. I believe this to be a solid submission and recommend it for acceptance.
> > >
> > > _Figure hyperrefs point to captions (below the figure)._ It's hard to provide helpful feedback on how to address this without looking at the authors' setup, but I've never had any problems following the simple instructions [here](https://www.overleaf.com/learn/latex/Cross%20referencing%20sections,%20equations%20and%20floats#Introduction).

---

### Official Review · AnonReviewer3 · 2020-10-30
**New realistic learning paradigm**

**Rating:** 6
**Confidence:** 3

**Review:**

Summary:

This work aims to make a realistic learning setting by combining few-shot learning and continual learning in the online setting. Similar to few-shot learning, the model needs to adapt to new classes with a few samples (at least in the beginning). Similar to continual learning, the model needs to learn new classes over time while being tested on the older classes as well. When encountering a new class, the model is expected to recognize that. Similar to the online setting, model evaluation happens on each trial, after which the model can be updated with that data (labeled or unlabeled). This new paradigm is called Online Contextualize Few-Shot Learning.

The authors recognize the importance of spatio-temporal context in human learning. Building on this, they propose a new dataset, RoamingRooms, that incorporates such context. The authors propose a new method, Contextual Prototypical Memory, to tackle this problem. It makes use of an RNN to encode contextual information and a prototype memory to remember previously learned classes.

Pros:
1. The new setting proposed, Online Contextualized Few-Shot Learning, is a very realistic setting. Few-shot learning misses that classes are repeated over time. Continual learning misses that new classes are not processed and learnt in groups. This new paradigm combines the two settings and improves on their shortcomings to make it more realistic. Additionally, this is all done in an online setting.
2. The use of spatio-temporal context in creating the dataset and the model is realistic and interesting.
3. The proposed model is simple, with components added specifically to make use of the additional information in the dataset (the RNN) or to output additional information required for the task (the new class detection branch).

Cons:
1. Authors mention that there exist some datasets under a very similar setting, namely CORe50, OpenLORIS, and synthetic task sequences of Omniglot and Tiered-ImageNet. If something similar does exist, the authors should report numbers on these datasets rather than RoamingOmniglot.
2. Authors mention that [1] proposed a model for a setting very similar to Online Contextualized Few-Shot Learning. Even if this method detects new classes by thresholding the probabilities for novel class detection, it should be used as a baseline method.
3. Clever fine-tuning is competitive, if not the state-of-the-art, for continual learning [2] and few-shot learning [3]. How does this perform as a baseline?
4. Average precision is used as the metric for this setting. What about the maximum F-1 scores?

Notes:
1. There is a lot going on in this paper. The writing can be made less redundant and more to the point to incorporate more details.

[1] Massimo Caccia et al. Online Fast Adaptation and Knowledge Accumulation: A New Approach to Continual Learning.
[2] Hang Qi et al. Low-Shot Learning with Imprinted Weights.
[3] Guneet S. Dhillon et al. A Baseline for Few-Shot Image Classification.

---

> ### Author Response · Authors · 2020-11-21
> **Response to Reviewer 3**
>
> Thank you for your comments and suggestions. Below are our responses.
>
> > Numbers on CORe50, OpenLORIS and synthetic task sequences of Omniglot and Tiered-ImageNet.
>
> Thank you for the suggestions. CORe50 contains only 50 object instances in total, so the maximum number of classes is 50 before train/val/test split; we worry that it is not enough for few-shot/meta-learning based approaches. As for OpenLORIS, it is not an incremental class learning benchmark. The continual variations across time are: Illumination, Occlusion, Object size, Camera-to-object angles/distances, Clutter. However, we will consider your suggestion of running on Tiered-ImageNet. Conceptually it will be similar to RoamingOmniglot, since in both cases, sequences are sampled from static images.
>
> > [1] proposed a model for a setting very similar to Online Contextualized Few-Shot Learning. Even if this method detects new classes by thresholding the probabilities for novel class detection, it should be used as a baseline method.
>
> We have studied [1] in detail and we think it cannot be applied in our setting. When we say similar, we mean both papers remove the notion of support+query split in each episode. However, there are still notable differences that prevent us from directly comparing with them. First, their setting is not incremental class learning, and instead of detecting when a class is novel, they detect when an environment is new based on the current training loss. Furthermore, in our setting, a model makes a class prediction before seeing the label; therefore it cannot use training loss to decide whether or not the current instance is a new class. Conceptually, the method proposed in [1] is a simple gradient based meta-learning method, and therefore we would like to refer the reviewer to the OML-U and OML-U++ results in our paper for a comparison for gradient-based meta-learners.
>
> > Clever fine-tuning is competitive, if not the state-of-the-art, for continual learning [2] and few-shot learning [3]. How does this perform as a baseline?
>
> We thank the reviewer for the suggestion. In our CPM model, fine-tuning the CNN could potentially corrupt old memory entries/prototypes and could introduce catastrophic forgetting. We can also try a pre-trained CNN + store all historical examples + fine-tuning. But we will still need to design a way for the model to predict known vs. unknown at each step, and this makes fine-tuning a less than straightforward baseline.

---

### Official Review · AnonReviewer4 · 2020-10-31
**An interesting and new learning setting**

**Rating:** 7
**Confidence:** 4

**Review:**

#################################

Summary:

The paper presented a new setting of online contextualized few shot learning to mimic human learning. This setting combines continual learning and few shot learning, and additionally considers context switch. Specifically, a learning method is presented with a sequence of samples that might come with labels. The method is then tasked to classify the current input into known categories, or recognize the input as belonging to a “new” category, while at the same time updating the model for known and new categories. Two new datasets (hand-written characters and indoor images) were constructed to support the learning setting. An extension of Prototypical Network (Snell et al.) was explored for this new setting. The results were compared against several baselines and were quite promising.

#################################

Pros:

* A novel setting of continual few-shot learning that considers context switching. The motivation is well articulated (naturalistic human learning). The setting has great potential to address some of the key challenges in AI (e.g., embodied vision, robotics, etc).
* New datasets to support the proposed setting. Those dataset might facilitate future research in this direction.
* The paper explored several baselines for the proposed setting, including an interesting extension of ProtoNet. The experiments are solid and the results are promising.

#################################

Cons:

* The evaluation metric will need some thoughts

Proper evaluation metric is a critical component for the proposed learning setting. While the authors did provide a short description of the evaluation metrics (AP and N-Shot accuracy), those metrics lack some details and are not well justified. My understanding is that both metrics are accumulated from the starting to the current step and across all sequences. This was not particularly clear from the text. Also the definition of AP is different from standard average precision (the TP/TN/FP/FN definitions are different). Some more description is needed in the text.

How do these metrics capture catastrophic forgetting? For example, the accuracy / AP vs. the time interval between the current label and the last time the same label was observed.


#################################

Minor comments:

Page 3 paragraph 2: “focuses on more flexible ...” not a sentence.

Table 1 and 2: Why the std of AP metric is not included?

Figure 5 caption does not match the layout. Is this 1-shot accuracy? I did not find the description in the paper.

#################################

Justification for score:

A good paper proposing an interesting learning paradigm. I’d expect some more discussion of the evaluation metric. Otherwise, I am happy to accept the paper.

---

> ### Author Response · Authors · 2020-11-21
> **Response to Reviewer 4 (Part 1)**
>
> Thank you for your comments and suggestions. Below are our responses.
>
> > (AP and N-Shot accuracy), those metrics lack some details and are not well justified. My understanding is that both metrics are accumulated from the starting to the current step and across all sequences. AP is different from standard average precision (the TP/TN/FP/FN definitions are different).
>
> We realize that our description of these metrics is not clear in the paper. We will update the paper.
>
> These metrics are justified by the domain. We chose to use AP (average precision or area under the precision-recall curve) as a way of integrating two aspects of performance:
>
> 1) the binary accuracy of whether an instance belongs to a known or unknown class (KU-Assign for short), and
> 2) the accuracy of assigning an instance the correct class label given it is from a known class (Class-Assign for short).
>
> The procedure to calculate AP is as follows. We first sort all the {KU-Assign, Class-Assign} predictions across all sequences in descending order based on KU-Assign probability, where the high ranked predictions should be known (not novel) classes. For the N top ranked instances in the sorted list, we compute:
>
> * precision@N = correct(Class-Assign)@N / N
> * recall@N = correct(Class-Assign)@N / K,
>
> where K is the true number of known instances and correct(Class-Assign)@N is the count of the number of correct class assignments among the top N. (The class assignment for an unknown instance is always incorrect.) To obtain the AP, we compute the integral of the function (y=precision@N, x=recall@N) across all N’s.
>
> N-shot accuracy: We define N-shot accuracy as the number of times an instance that has been seen N times thus far in the sequence is classified correctly. We compute the mean and standard error of this over all sequences.
>
> > How do these metrics capture catastrophic forgetting? For example, the accuracy / AP vs. the time interval between the current label and the last time the same label was observed.
>
> It is true that these metrics do not explicitly capture catastrophic forgetting (CF). In this setting it is difficult to pull out CF explicitly because of the increase in the number of classes over time. So a decrease in accuracy could be due to both CF and the increasing difficulty based on the number of classes. We provide two different measurements, and both are unfortunately coupled with the total number of classes.
>
> First, we would like to point the reviewer to Figure 5, where we show instantaneous accuracy over time, which drops as the number of classes increases.
>
> Second, we also plot a 2D table here in our rebuttal. On the y-axis is the number of times we have seen item X’s label for K times (K-shot), and on the x-axis we look at the time interval since we last saw such item X. We are comparing CPM with Online ProtoNet (OPN)
>
> **RoamingOmniglot (Last seen interval vs. K-shot)**
>
> |     **Interval** | **1 ~ 2** | **3 ~ 5** | **6 ~ 10** | **11 ~ 20** | **21 ~ 50** | **51 ~ 100** |
> |--------------|-------|-------|--------|---------|---------|----------|
> | OPN 1-shot   | 88.8  | 86.9  | 85.2   | 84.7    | 83.6    | 81.1     |
> | CPM 1-shot   | **96.06** | **94.01** | **92.95**  | **91.56**   | **88.21**   | **84.58**    |
> | OPN 3-shot   | 97.2  | 97.1  | 96.6   | 96.7    | **96.5**    | 95.3     |
> | CPM 3-shot   | **98.48** | **98.16** | **97.46**  | **97.17**   | 95.37   | **95.53**    |
>
> **RoamingOmniglot-Semi-Sup**
>
> |     **Interval** | **1 ~ 2** | **3 ~ 5** | **6 ~ 10** | **11 ~ 20** | **21 ~ 50** | **51 ~ 100** |
> |------------|-------|-------|--------|---------|---------|----------|
> | OPN 1-shot | 90.05 | 88.93 | 88.41  | 87.60   | 87.31   | 85.12    |
> | CPM 1-shot | **95.86** | **93.84** | **92.81**  | **91.78**   | **89.36**   | **85.68**    |
> | OPN 3-shot | 97.77 | 97.33 | 97.11  | **97.75**   | **97.69**   | **96.78**    |
> | CPM 3-shot | **98.71** | **97.53** | **97.54**  | 96.51   | 96.34   | 92.93    |
>
> **RoamingRooms**
>
> |     **Interval** | **1 ~ 2** | **3 ~ 5** | **6 ~ 10** | **11 ~ 20** | **21 ~ 50** | **51 ~ 100** |
> |------------|-------|-------|--------|---------|---------|----------|
> | OPN        | 93.47 | 89.30 | 79.35  | 67.19   | 60.27   | 60.06    |
> | CPM 1-shot | **95.70** | **92.16** | **85.68**  | **75.15**   | **70.04**   | **66.36**    |
> | OPN 3-shot | 95.11 | 91.80 | 85.55  | 78.19   | 74.64   | 73.84    |
> | CPM 3-shot | **96.14** | **93.78** | **87.73**  | **81.40**   | **79.09**   | **78.20**    |

---

> > ### Author Response · Authors · 2020-11-21
> > **Response to Reviewer 4 (Part 2)**
> >
> > **RoamingRooms-Semi-Sup**
> >
> > |     **Interval** | **1 ~ 2** | **3 ~ 5** | **6 ~ 10** | **11 ~ 20** | **21 ~ 50** | **51 ~ 100** |
> > |------------|-------|-------|--------|---------|---------|----------|
> > | OPN 1-shot | 86.50 | 83.59 | 76.33  | 68.39   | 64.69   | 61.48    |
> > | CPM 1-shot | **91.01** | **88.73** | **82.89**  | **76.99**   | **72.16**   | **66.50**    |
> > | OPN 3-shot | 92.60 | 88.03 | 85.08  | 81.07   | 80.64   | 76.67    |
> > | CPM 3-shot | **94.82** | **91.04** | **86.89**  | **83.09**   | **82.72**   | **79.17**    |
> >
> > As shown above, there is a drop in accuracy if we haven’t seen an item for a long time. However, the time interval is again coupled with the increase of total number of classes. The amount of drop in accuracy is smaller for 3-shot accuracy than 1-shot. In RoamingRooms the drop seems to be larger than RoamingOmniglot, but this might also be due to 1) the training sequences are shorter (100 vs. 150), and 2) the context switching happens less often during training than RoamingOmniglot. In general, we don’t think there is severe catastrophic forgetting in our online few-shot learning experiments. Catastrophic forgetting usually happens when a network adapts the weights and representation. In our model, CPM, the CNN is frozen during test episodes. In theory, CPM could still lead to forgetting due to RNN activation and Gated Averaging Unit, but this is rarely the case as shown in the results above, where it only loses to Online ProtoNet a few times, and wins most of the time due to its ability to model temporal contexts. Therefore, we believe that the drop in accuracy was mostly due to the increase in the total number of classes.
> >
> > We will include these additional results in our next version of the paper.
> >
> > > Table 1 and 2: Why the std of AP metric is not included?
> >
> > As explained above, the AP metric is calculated by aggregating all the predictions from all sequences together, so that we can rank all the scores together. And therefore, there is no std for the AP metric. But since we are aggregating 2000 sequences together, and each sequence has around 100 items, the score is very stable and does not fluctuate much.
> >
> > > Figure 5 caption does not match the layout. Is this 1-shot accuracy? I did not find the description in the paper.
> >
> > The standard k-shot N-way few-shot accuracy does not fit neatly into this online setting, and therefore we measure the instantaneous accuracy at each time step, aggregated over 2000 sequences.

---

> > > ### Comment · AnonReviewer4 · 2020-11-24
> > > **Updated Review**
> > >
> > > I appreciate the authors' effort for clarifying the evaluation metrics. Now these metrics make more sense to me.
> > >
> > > Overall, I think this is a good paper and would recommend its acceptance.

---

### Comment · ~Yuliang_Zou1 · 2021-02-04
**Evaluation?**

Thanks for the great work! I feel this paper very interesting.

However, I am a bit confused about the evaluation (or dataset split). According to Figure 1, the model always evaluates its prediction first when given a new example, and then update/train itself. And as the caption says, "there is no separate testing phase; model training and evaluation happen at the same time". So I wonder why the datasets are split as train/val/test and how to use them.

---

> ### Comment · ~Mengye_Ren1 · 2021-02-04
> **Re: Evaluation**
>
> Dear Yuliang,
>
> Thank you for your interest in our paper. Here the sequence is the concept of an "episode" in standard FSL. Therefore, the full evaluation/meta-test will run on 2000 sequences, and each sequence consists of a stream of images. And the meta-training is run on many sequences/episodes as well. When we wrote "there is no separate testing phase," we meant there is no notion of "support set" or "query set" in our sequence. Everything in the sequence is both a "support" and a "query".
>
> For dataset split, on Omniglot we follow the standard class-wise/alphabet split so that the test classes are not seen during training. For matterport, we split 60/10/20 home environments for train/val/test.
>
> Evaluation metrics: We chose to use AP (average precision or area under the precision-recall curve) as a way of integrating two aspects of performance:
> 1) the binary accuracy of whether an instance belongs to a known or unknown class (KU-Assign for short), and
> 2) the accuracy of assigning an instance the correct class label given it is from a known class (Class-Assign for short).
> The procedure to calculate AP is as follows. We first sort all the {KU-Assign, Class-Assign} predictions across all sequences in descending order based on KU-Assign probability, where the high ranked predictions should be known (not novel) classes. For the N top ranked instances in the sorted list, we compute:
>
> precision@N = correct(Class-Assign)@N / N
> recall@N = correct(Class-Assign)@N / K,
> where K is the true number of known instances and correct(Class-Assign)@N is the count of the number of correct class assignments among the top N. (The class assignment for an unknown instance is always incorrect.) To obtain the AP, we compute the integral of the function (y=precision@N, x=recall@N) across all N’s.
>
> N-shot accuracy: We define N-shot accuracy as the number of times an instance that has been seen N times thus far in the sequence is classified correctly. We compute the mean and standard error of this over all sequences.
>
> I hope this answers your question.

---

> > ### Comment · ~Yuliang_Zou1 · 2021-02-04
> > **RE**
> >
> > I see. So you train your model how to use prototypical memory to do matching and novelty detection on the training set. And then you reset the memory to zero and run your model without network parameter updating/training on the test set to do the evaluation.

---

> > > ### Comment · ~Mengye_Ren1 · 2021-02-05
> > > **Re**
> > >
> > > Yes, what you describe is correct.

---

### Decision · Program_Chairs · 2021-01-07
**Final Decision**

**Decision:**

Accept (Poster)

**Comment:**

This paper proposes a new online contextualized few-shot learning setting, with two associated datasets (notably, including one obtained from trajectories within the real-world Matterport3D reconstructions). A simple recurrent contextualized extension of Prototypical Networks is also proposed as a stronger baseline, demonstrating the need for incorporating such context. The reviewers all agreed that this is an interesting setting combining continual and few-shot learning, offering a more realistic problem that mirrors those that might be encountered by embodied agents. The authors provided very detailed rebuttals, answering some of the questions and concerns raised by the reviewers. In the end, all reviewers agreed that this paper would contribute a significant novel setting, and so I recommend acceptance. I encourage the others to include modifications related to some of the comments, such as strengthening/clarifying the setting including metrics, details of the method, etc.